# Exact first-order effect of interactions on the ground-state energy of harmonically-confined fermions

Pierre Le Doussal[1], Naftali R. Smith[2*] and Nathan Argaman[3,4]

**1** Laboratoire de Physique de l'Ecole Normale Supérieure, CNRS, ENS & Université PSL, Sorbonne Université, Université Paris Cité, 24 rue Lhomond, 75005 Paris, France
**2** Department of Environmental Physics, Blaustein Institutes for Desert Research, Ben-Gurion University of the Negev, Sede Boqer Campus, 8499000, Israel
**3** Department of Physics, Ben-Gurion University of the Negev, Be'er-Sheva 84105, Israel
**4** Department of Physics, Nuclear Research Center—Negev, P.O. Box 9001, Be'er Sheva 84190, Israel

★ naftalismith@gmail.com

## Abstract

We consider a system of $N$ spinless fermions, interacting with each other via a power-law interaction $\epsilon/r^n$, and trapped in an external harmonic potential $V(r) = r^2/2$, in $d = 1, 2, 3$ dimensions. For any $0 < n < d + 2$, we obtain the ground-state energy $E_N$ of the system perturbatively in $\epsilon$, $E_N = E_N^{(0)} + \epsilon E_N^{(1)} + O\left(\epsilon^2\right)$. We calculate $E_N^{(1)}$ exactly, assuming that $N$ is such that the "outer shell" is filled. For the case of $n = 1$ (corresponding to a Coulomb interaction for $d = 3$), we extract the $N \gg 1$ behavior of $E_N^{(1)}$, focusing on the corrections to the exchange term with respect to the leading-order term that is predicted from the local density approximation applied to the Thomas-Fermi approximate density distribution. The leading correction contains a logarithmic divergence, and is of particular importance in the context of density functional theory. We also study the effect of the interactions on the fermions' spatial density. Finally, we find that our result for $E_N^{(1)}$ significantly simplifies in the case where $n$ is even.



# 1  Introduction

## 1.1  Fermions in traps

The spectacular experimental developments in manipulating cold atoms (bosons or fermions) [1,2], allow to probe in great detail the quantum many-body physics, both for interacting and noninteracting systems. Cold gases display nontrivial behavior even in the zero-temperature limit, due to the quantum nature of the particles [3–5]. These properties are experimentally accessible, for instance using Fermi quantum microscopes [6–8]. The experiments often involve inhomogeneous environments, such as optical traps, and this has led to a renewed theoretical interest in the problem of fermions and bosons in confining external potentials. In these systems, not only a large variety of confining potentials can be realized [1,6–10], but the

nature and the strength of the interactions between the particles can also be tuned [1,9]. In particular, there is a lot of interest recently in long-range interactions, including experimental realizations [11]. Moreover, the non-interacting limit can be reached. The idealized model of noninteracting fermions in a trap has thus been reexamined. In a confining potential, the Fermi gas is supported over a finite domain and its mean density is not, in general, spatially uniform. Due to the Pauli principle and the inhomogeneous setting, it exhibits non trivial spatio-temporal quantum correlations. The simplest approximation based on free fermions with a slowly varying density fails near the edge of the gas, where the density vanishes. More elaborate methods have been developed to handle non-interacting fermions in inhomogeneous environments, such as the inhomogeneous bosonisation [12], mostly to describe the bulk of the gas, or exact methods which can also describe the edge [13–15], based on the theory of determinantal point processes, much developed in random matrix theory [16–18]. In fact, in some favorable cases in one space dimension, $d = 1$, or in some particular situations in $d = 2$, exact mappings to random matrices exist [19–21, 23, 24], which lead to exact solutions.

By contrast, the case of interacting fermions in a confining potential is far more difficult to tackle. A fundamental question, which will be our main focus here, is to determine the many-body ground-state energy: In the noninteracting case, it is simply obtained as the sum of single-body energy levels associated to the external potential. In the interacting case even addressing that basic question is quite difficult, and studying other properties such as the density of the gas is even more challenging. For a large number $N \gg 1$ of fermions, both the ground-state energy and the density of the gas can be obtained, in the leading-order, using the celebrated Thomas-Fermi approximation. In some extremely special cases, the many-body Schrödinger problem in presence of an external trap remains integrable in presence of interactions, and all eigenenergies can be found exactly. This is the case for instance for the Calogero model, which describes $N$ spinless fermions in a harmonic trap in dimension $d = 1$ with inverse-square interactions [25,26]. Remarkably, this is also a case where a mapping to random matrices exists in presence of interactions, i.e. the joint distribution of the positions of the fermions coincides, up to scaling factors, with that of the eigenvalues of an $N \times N$ random matrix sampled from the Gaussian $\beta$ ensemble [27]. However, there is a considerable gap in the literature regarding the extension of these results, obtained in special cases, to more general settings (i.e., general trapping potentials, interactions, space dimensions and nonzero temperature).

## 1.2 An example: the atom

One important example of fermions in an external potential is the case of the electrons in an atom. In the early eighties [29,30], Schwinger applied semiclassical methods to the ground-state energy of neutral atoms, disregarding relativistic effects, and found that [31]

$$E \simeq -0.768\,745\,Z^{7/3} + \tfrac{1}{2}Z^2 - 0.269\,900\,Z^{5/3}, \tag{1}$$

where $Z$ is the atomic number (Hartree atomic units are used; see [32] for a mathematical proof). The leading term here results from the Thomas-Fermi approximation, which balances the effects of the nuclear attraction, the kinetic energy of the electrons (to leading order), and their mutual repulsion, treated at the Hartree level. The next term, the Scott correction [29], arises from the quantization of the deepest energy levels. The $Z^{5/3}$ term is where the exchange energy begins to contribute [30] (in fact, 9/11 of this term is due to the exchange energy, $E_x$, with the remaining 2/11 due to corrections to the kinetic energy; correlations only contribute to the $Z \ln Z$ and higher order terms not shown in (1), see below). Schwinger noted the "unreasonable utility" of such asymptotic expansions [29]: the expression above, despite being derived for $Z \to \infty$, is accurate to better than 1% for all $Z > 5$, and better than 10% for *all* $Z$, down to $Z = 1$. Indeed, the coefficients of the next term in the series, which is much more

involved as it describes oscillations in $Z$, are more than an order of magnitude smaller than the leading coefficient above [31].

A significantly higher accuracy is achieved with the density functional theory (DFT), even when applied using its most basic version, the so-called local density approximation (LDA) [33]. However, typical applications, such as the binding energy of a molecule — the difference between its ground-state energy and that of the separate atoms — require still higher accuracy, and the LDA has been supplemented by various additional approximate terms, achieving considerable improvements. Understanding and gauging the accuracy of the different approximations can be difficult, as they often involve uncontrolled approximations. Only fairly recently, in 2009 [34], the use of an asymptotic expansion for the *inaccuracy* of the LDA was suggested. This inaccuracy is dominated by its exchange part [35], $\Delta E_\mathrm{x} = E_\mathrm{x} - E_\mathrm{x}^{\mathrm{LDA}}$. At first, an expansion in powers of $Z^{1/3}$ was sought, but this expansion in fact begins with a logarithmic term [36, 37],

$$\Delta E_\mathrm{x} \simeq -BZ \ln Z - CZ \tag{2}$$

(this expression too is accurate to better than 10% for all $Z$). The values of the coefficients are $B = 1/(4\pi^2)$ and $C \simeq 0.056$. A comparison of this with Schwinger's expression above provides a very clear specification of the accuracy of the LDA for total energies, identifying a major challenge for improved DFT approximations.

Despite the fact that $B$ is known precisely, it remains enigmatic. It was originally expected that corrections to the LDA would follow from an expansion in weak gradients [33], and the leading coefficient of the gradient expansion approximation, $\mu^{\mathrm{GE}} = 10/81$, was carefully derived by applying perturbation theory to the homogeneous electron gas [38] (here the perturbation is an inhomogeneous external potential, not the interaction). This gives the correct qualitative result, but is quantitatively wrong. The actual value of $B$ is known only from numerical studies [36]. One such study was performed for the Bohr atom — a system of noninteracting electrons moving in the Coulomb potential of a nucleus, and thus having analytically known hydrogenic wavefunctions. Considering the interaction between electrons perturbatively to leading order, accurate values of the exchange energy could be obtained analytically for "atoms" with very large numbers of electrons. Fitting these to an asymptotic expansion yielded coefficients with several-digit accuracy, allowing one to guess at the exact values of the coefficient $B$ of the $Z \ln Z$ term. Confirmation of these values was obtained by fixing the value of $B$ and observing how much easier it became to fit the remaining higher-order coefficients in the expansion. For the Bohr atom, the overall result is $B_o = 1/(3\pi^2)$.

The gradient expansion indicates that there are two logarithmic contributions for the Bohr atom: one from a $\int dr/r$ integration over the region $Z^{-1} \ll r \ll Z^{-1/3}$, and another from a $\int dr/(r_c - r)$ integration over the region $Z^{-5/9} \ll r_c - r \ll Z^{-1/3}$, where $r_c = (18/Z)^{1/3}$ is the radius at the edge of the electron distribution, where the chemical potential is equal to the nuclear potential (both the very inner and the very outer limits are determined by the wavelength of the electrons at the Fermi level; $Z^{-1/3}$ is the scale of the overall density distribution). The second contribution is three times smaller than the first, and is absent from real atoms, for which screening drastically reduces the electric field at the edge of the electron distribution, explaining the difference between $B$ and $B_o$. Note that the homogeneous electron gas, and any distribution obtainable from it by the weak perturbations considered in Ref. [38], possess neither a divergent potential, which leads to the inner logarithmic integration, nor a sharp edge, which leads to the outer one. Thus, a quantitative explanation of the values of $B$ and $B_o$ is still lacking.

Note that for heavy, neutral atoms, the atomic number $Z$ plays a double role: On the one hand, it is the number of fermions (electrons). On the other hand, $Z^{-1}$ is the ratio between the strength of the electron-electron interactions and the electron-nucleus interactions. As explained below, in much of the remainder of this paper we will assume that the number of

fermions $N$ is large and the strength $\epsilon$ of the interactions between them is weak, but we will not assume a connection between the two parameters $N$ and $\epsilon$.

As DFT is the method of choice for treating many-electron systems (see, e.g., [39]), and is in very wide use (scores of thousands of publications per year), the above gives ample motivation to study additional systems and to focus on the leading corrections to the exchange energy, seeking logarithmic contributions in particular. A specific example will be provided below, and the implications for DFT will be studied and reported separately [40].

### 1.3 Aim of the present work, and outline

In this paper we provide a significant step in the direction of understanding the general quantitative behavior of the ground state energy of fermions in external potentials. To obtain analytical results we consider here only the case of the harmonic trap, but we are able to treat general power-law interactions in arbitrary space dimensions. We focus here on the expression for the ground state energy which is predicted to first order in perturbation theory in the interactions, i.e. we assume that these are weak ($\sim \epsilon$). However, for this first order prediction, we obtain the full exact result, valid for any number of fermions $N$, corresponding to a filled highest energy shell. This is achieved using methods of determinantal point processes. We then study the large-$N$ behavior and from the exact result we obtain the corresponding series expansions in powers of $N$ to a high order. These series turn out to be numerically very accurate, and they contain interesting logarithmic terms. There is much to learn from finding interpretation for these terms. Here we only discuss the leading term; the identification of the semiclassical expression for the leading logarithmic correction will be studied in a separate publication [40].

The remainder of the paper is organized as follows. In Section 2 we precisely define the model under study and give a summary of our main findings. In Section 3 we perform the exact calculation of the ground-state energy, to leading order in perturbation theory with respect to the interaction strength. We then study the $N \gg 1$ asymptotic behavior. In Section 4 we obtain the leading-order effect of the interactions at $N \gg 1$ using semiclassics. In Section 5, for the sake of completeness, we calculate the leading-order effect of the interactions on the gas density at $N \gg 1$ using the Thomas-Fermi approximation. In Section 6 we obtain explicit exact results for some special cases of the interaction, where certain simplifications occur: these recover, and extend (to first order in the interaction) the result for the Calogero model. In Section 7 we summarize and discuss our results. Some of the more technical calculations are given in the appendices.

## 2 Model, definitions and summary of main results

The system that we study consists of $N$ identical, trapped interacting spinless fermions of unit mass in $d$ dimensions (the effects of spin degeneracy $g_0$ are straightforward to incorporate if necessary). The Hamiltonian of the system is

$$\hat{H} = \sum_{i=1}^{N} \left( \frac{p_i^2}{2} + V(x_i) \right) + \epsilon \sum_{1 \le i < j \le N} W(x_i, x_j), \tag{3}$$

where $V(x)$ is the trapping potential and $W(x, y)$ is the interaction term. Our goal is to perturbatively study the effect of the interaction on the system. In most of what follows, we consider the case of the harmonic trapping potential for which analytical results can be obtained. Although we derive an intermediate formula valid for general interaction $W(x, y)$,

our explicit results are obtained for the case where the interaction is a decaying power law, i.e., we focus on

$$V(x) = \frac{x^2}{2}, \qquad W(x, y) = |x - y|^{-n},\tag{4}$$

where we have chosen units such that $\hbar$ and the stiffness of the potential are both equal to unity, and we denote the modulus of the vector $x$ by $x$. Our main goal is to calculate the ground-state energy of this many-body system, which we achieve in the weakly-interacting (i.e., small-$\epsilon$) limit. We begin by analyzing the system in the absence of interactions, $\epsilon = 0$ and then apply first order perturbation theory to calculate the leading-order correction in $\epsilon$.

## 2.1 Noninteracting case

In the noninteracting case, $\epsilon = 0$, the single-body energy levels of the harmonic potential are given by $\mathcal{E}_{k_1,\dots,k_d} = k_1 + \cdots + k_d + \frac{d}{2}$, where $k_i = 0, 1, 2, \dots$. The many-body ground state is obtained by filling up the $N$ lowest single energy levels, so that the ground-state energy $E_N^{(0)}$ is straightforward to compute. We denote the Fermi energy $\mu$ as the energy of the highest occupied level,

$$\mu = M - 1 + \frac{d}{2},\tag{5}$$

where $M = 1, 2, \dots$. In this paper we only consider the case where the highest occupied level is fully occupied so that the many-body ground state of the non-interacting problem is non-degenerate (filled shell). This restricts the allowed values for $N$. By counting single-body energy levels one finds that [41]

$$N = \sum_{k=1}^{M} \binom{k+d-2}{d-1} = \binom{M+d-1}{d} = \begin{cases} M, & d = 1, \\ \frac{M(M+1)}{2}, & d = 2, \\ \frac{M(M+1)(M+2)}{6}, & d = 3. \end{cases}\tag{6}$$

On the other hand, the many-body ground state energy is the sum of the individual energy levels so[1]

$$\begin{aligned} E_N^{(0)} &= \sum_{k=1}^{M} \left(k - 1 + \frac{d}{2}\right)\binom{k+d-2}{d-1} \\ &= \frac{M(2M+d-1)}{2(d+1)}\binom{M+d-1}{d-1} = \begin{cases} \frac{M^2}{2}, & d = 1, \\ \frac{M(M+1)(2M+1)}{6}, & d = 2, \\ \frac{M(M+1)^2(M+2)}{8}, & d = 3. \end{cases} \end{aligned}\tag{7}$$

Eqs. (6) and (7) give $E_N^{(0)}$ as a function of $N$.

The ground state wave function $\Psi_0(x_1, \cdots, x_N)$ is also straightforward to find. It is given by the $N \times N$ Slater determinant constructed from the $N$ lowest single-body energy wave functions,

$$\Psi_0(x_1, \cdots, x_N) = \frac{1}{\sqrt{N!}} \det_{1 \le i, j \le N} \psi_i(x_j).\tag{8}$$

The latter (after relabeling the indices, $i \to k_1, \dots, k_d$) are given by

$$\psi_{k_1,\dots,k_d}(x) = \prod_{j=1}^{d} e^{-x_j^2/2} \left(\frac{1}{\sqrt{\pi} \, 2^{k_j} k_j!}\right)^{1/2} H_{k_j}(x_j)\tag{9}$$

---

[1]The combinatorial identity used in Eq. (7), i.e., moving from the first line of the equation to the second, is easy to prove by induction on $M$.

with $0 \leq k_1 + \cdots + k_d \leq M - 1$, where $H_i$ is the $i$th Hermite polynomial and we denote $\boldsymbol{x} = (x_1, \ldots, x_d)$.

Much is known about the spatial properties of trapped noninteracting fermions, see e.g. [15] for details and derivations. We now recall some of these properties which will prove useful to us later when we treat the interacting case. One can write the joint PDF of the fermions' positions as a single $N \times N$ determinant[2]

$$|\Psi_0(\boldsymbol{x}_1, \cdots, \boldsymbol{x}_N)|^2 = \frac{1}{N!} \det_{1 \leq i,j \leq N} K_N(\boldsymbol{x}_i, \boldsymbol{x}_j) \tag{10}$$

of a matrix whose entries are given by the so-called kernel

$$K_N(\boldsymbol{x}, \boldsymbol{y}) = \sum_{i=1}^{N} \psi_i^*(\boldsymbol{x}) \psi_i(\boldsymbol{y}). \tag{11}$$

This, together with the "reproducing" property of the kernel

$$\int K_N(\boldsymbol{x}, \boldsymbol{y}) K_N(\boldsymbol{y}, \boldsymbol{z}) d\boldsymbol{y} = K_N(\boldsymbol{x}, \boldsymbol{z}) \tag{12}$$

makes the joint PDF of $\boldsymbol{x}_1, \cdots, \boldsymbol{x}_N$ a determinantal point process [15,42,43]. Here and below $\int d\boldsymbol{y}$ denotes the $d$-dimensional integral over $\mathbb{R}^d$. As a result, one can express the $k-$point correlation function

$$R_k(\boldsymbol{x}_1, \cdots, \boldsymbol{x}_k) = \frac{N!}{(N-k)!} \int d\boldsymbol{x}_{k+1} \cdots d\boldsymbol{x}_N |\Psi_0(\boldsymbol{x}_1, \cdots, \boldsymbol{x}_N)|^2 \tag{13}$$

as a $k \times k$ determinant

$$R_k(\boldsymbol{x}_1, \cdots, \boldsymbol{x}_k) = \det_{1 \leq i,j \leq k} K_N(\boldsymbol{x}_i, \boldsymbol{x}_j). \tag{14}$$

This property enables one to calculate spatial properties of the fermions directly from the kernel. Consider, for instance, fermions' number density

$$R_1(\boldsymbol{x}) = N\rho_N(\boldsymbol{x}) = \left\langle \sum_{i=1}^{N} \delta(\boldsymbol{x} - \boldsymbol{x}_i) \right\rangle_0, \tag{15}$$

where $\langle \cdots \rangle_0$ denotes the expectation value with respect to the ground state $\Psi_0$. Note that the density is normalized such that $\int N\rho_N(\boldsymbol{x}) d\boldsymbol{x} = N$. Then the density is given, due to (14) with $k = 1$, by

$$N\rho_N(\boldsymbol{x}) = K_N(\boldsymbol{x}, \boldsymbol{x}). \tag{16}$$

Similarly, for $k = 2$, (14) gives the two-point function

$$R_2(\boldsymbol{x}, \boldsymbol{y}) = \left\langle \sum_{1 \leq i \neq j \leq N} \delta(\boldsymbol{x} - \boldsymbol{x}_i)\delta(\boldsymbol{y} - \boldsymbol{x}_j) \right\rangle_0 = K_N(\boldsymbol{x}, \boldsymbol{x}) K_N(\boldsymbol{y}, \boldsymbol{y}) - K_N(\boldsymbol{x}, \boldsymbol{y}) K_N(\boldsymbol{y}, \boldsymbol{x}). \tag{17}$$

---

[2]Note that, since the eigenfunctions (9) are real, the absolute value in Eq. (10) and the complex conjugate in Eq. (11) are in fact unnecessary for the particular case treated in the present work.

## 2.2 Interacting case and summary of main results

For nonzero interaction $\epsilon > 0$, the problem becomes significantly harder to solve. However, in the limit $\epsilon \to 0$, one can apply regular perturbation theory to obtain the expansion $E_N = E_N^{(0)} + \epsilon E_N^{(1)} + O\left(\epsilon^2\right)$ of the many-body ground state energy. $E_N^{(1)}$ is given by the expectation value of the interaction term in the Hamiltonian in the unperturbed ground state:

$$E_N^{(1)} = \left\langle \sum_{1 \le i < j \le N} W\left(x_i, x_j\right) \right\rangle_0 . \tag{18}$$

Now, rewriting this in the form[3]

$$E_N^{(1)} = \frac{1}{2} \int dx dy R_2\left(x, y\right) W\left(x, y\right) \tag{19}$$

and then using Eq. (17), we reach

$$E_N^{(1)} = \frac{1}{2} \int dx dy W\left(x, y\right) \left[K_N\left(x, x\right) K_N\left(y, y\right) - K_N\left(x, y\right) K_N\left(y, x\right)\right]. \tag{20}$$

One can separate this expression into two terms

$$E_N^{(1)} = \left(F_N - G_N\right)/2, \tag{21}$$

where

$$F_N = \int dx dy K_N\left(x, x\right) K_N\left(y, y\right) W\left(x, y\right) \tag{22}$$

and

$$G_N = \int dx dy K_N\left(x, y\right) K_N\left(y, x\right) W\left(x, y\right) \tag{23}$$

are the direct and exchange terms, respectively, provided that each of the two integrals (22) and (23) converges. As we will show below, in the case of the pure power law interaction $W(x, y) = |x - y|^{-n}$ the integral (20) converges for $n < d + 2$, while the integrals (22) and (23) converge separately provided the stronger condition $n < d$ holds. These divergences signal a breakdown of perturbation theory and can be cured by adding a small scale cutoff to the interaction, an extension not studied here. Below we restrict to the case $n < d + 2$.

Our main results are as follows. We calculate $E_N^{(1)}$ exactly for any $N$ for which the energy shells are all full (see above), and for power law interaction $W(x, y) = |x - y|^{-n}$ for any $0 < n < d + 2$ (note that $n$ can be a real number). The explicit formula are given in Eqs. (52) and (53) below, as well as in Eqs. (55) and (56), where $M$ is related to $N$ by Eq. (6), and $E_N^{(1)} = \frac{1}{2}(F_M - G_M)$.

For $n = 1$, which we refer to hereafter as the Coulomb interaction (since it indeed corresponds to the electrostatic interaction in $d = 3$ and also in lower-dimensional systems embedded in three-dimensional space), we analyze the asymptotic behavior of $E_N^{(1)}$ at $N \gg 1$. In $d = 1$, we find that

$$E_N^{(1)} \simeq \frac{8\sqrt{2} N^{3/2} \left(3 \ln N + 3\gamma - 14 + 18 \ln 2\right)}{9\pi^2}, \tag{24}$$

---

[3]The factor $1/2$ in Eq. (19) is there because in Eq. (17) the sum is over $i \ne j$, whereas in Eq. (18) it is over $i < j$.

where $\gamma = 0.577\ldots$ is the Euler constant. For $d > 1$ we can analyze the direct and exchange terms separately (because they do not diverge). In $d = 2$, we obtain

$$F_N \simeq \frac{1024 \times 2^{1/4} N^{7/4}}{315\pi} + \frac{2^{5/4} N^{3/4}}{45\pi}, \tag{25}$$

$$G_N \simeq \frac{1}{\pi\sqrt{2}} \left[ \frac{64 \times 2^{1/4}}{15} N^{5/4} + \frac{3\ln(2N) + 6c_2 - 8}{12 \times 2^{3/4}} N^{1/4} \right], \tag{26}$$

and in $d = 3$ we find

$$F_N \simeq \frac{131072 \times 2^{1/3} N^{11/6}}{17325 \times 3^{1/6}\pi^2} - \frac{128 \times 2^{2/3} N^{7/6}}{945 \times 3^{5/6}\pi^2} + \frac{67\sqrt{N}}{2100\sqrt{3}\pi^2}, \tag{27}$$

$$G_N \simeq \frac{2}{\pi^2} \left[ \frac{64 \times 2^{2/3} 3^{1/6}}{35} N^{7/6} + \frac{5\ln(6N) + 15c_3 - 176}{30\sqrt{3}} \sqrt{N} \right], \tag{28}$$

where

$$c_2 = 6\ln 2 + \gamma - \frac{13}{6}, \quad c_3 = 6\ln 2 + \gamma + \frac{47}{6}. \tag{29}$$

Notable in Eqs. (24), (26) and (28) are the terms with $\ln N$ in the expansions, on which we will comment in further detail below. The above formula are valid up to corrections which decay at large $N$, and the above series are displayed (as a function of $N$ and of $M$) with higher accuracy in Section 3 below.

We also obtained explicit exact results for other integer values $n > 1$, as well as their corresponding large-$N$ behaviors to high accuracy. The case $n = d$ is technically delicate, and is treated in Appendix B, with explicit results for the cases $n = d = 1, 2, 3$. For even $n$, certain simplifications arise; We obtain explicit results for the cases $(d = 1, n = 2)$, $(d = 3, n = 2)$, $(d = 3, n = 4)$ and $(d = 2, n = 3)$ in section 6. Finally, we have obtained an intermediate formula (41) valid for a larger class of interactions, which allows in principle to analyze also these cases (not performed here).

# 3 Ground-state energy in the weakly interacting case: Exact results and large-$N$ asymptotic behavior

## 3.1 General exact formula for $E_N^{(1)}$

To compute the correction $E_N^{(1)}$ to the ground state energy in (20) for harmonic confinement, we will use an exact formula for a generating function of the kernel, denoted $\mathsf{K}_z(\boldsymbol{x}, \boldsymbol{y})$ below, which allows to conveniently perform the spatial integration in any dimension, for any $n$. The formula is a generalization of Mehler's formula and is given in Eq. (2) in [44]. It reads (in our notation)

$$\mathsf{K}_z(\boldsymbol{x}, \boldsymbol{y}) = \sum_{M=1}^{\infty} z^M \mathcal{K}_M(\boldsymbol{x}, \boldsymbol{y})$$

$$= \frac{z}{1-z} \frac{1}{\pi^{d/2}(1-z^2)^{d/2}} \exp\left( \frac{4z\,\boldsymbol{x} \cdot \boldsymbol{y} - (1+z^2)(x^2 + y^2)}{2(1-z^2)} \right), \tag{30}$$

where we are now using $M$ for the label of the kernel, i.e. we denote from now on

$$\mathcal{K}_M(\boldsymbol{x}, \boldsymbol{y}) = K_N(\boldsymbol{x}, \boldsymbol{y}), \tag{31}$$

where $N$ and $M$ are related by Eq. (6).

Let us consider a translationally invariant interaction $W(\boldsymbol{x}, \boldsymbol{y}) = W(\boldsymbol{x} - \boldsymbol{y})$, and define the spatial integrals, for $M_1, M_2 \geq 1$

$$Q_{M_1, M_2} = \int d\boldsymbol{x} d\boldsymbol{y} W(\boldsymbol{x} - \boldsymbol{y}) \left[ \mathcal{K}_{M_1}(\boldsymbol{x}, \boldsymbol{x}) \mathcal{K}_{M_2}(\boldsymbol{y}, \boldsymbol{y}) - \mathcal{K}_{M_1}(\boldsymbol{x}, \boldsymbol{y}) \mathcal{K}_{M_2}(\boldsymbol{y}, \boldsymbol{x}) \right], \quad (32)$$

as well as its generating function

$$
\begin{aligned}
Q(z_1, z_2) &= \sum_{M_1, M_2 \geq 1} Q_{M_1, M_2} z_1^{M_1} z_2^{M_2} \\
&= \int d\boldsymbol{x} d\boldsymbol{y} W(\boldsymbol{x} - \boldsymbol{y}) \left[ \mathsf{K}_{z_1}(\boldsymbol{x}, \boldsymbol{x}) \mathsf{K}_{z_2}(\boldsymbol{y}, \boldsymbol{y}) - \mathsf{K}_{z_1}(\boldsymbol{x}, \boldsymbol{y}) \mathsf{K}_{z_2}(\boldsymbol{y}, \boldsymbol{x}) \right], \quad (33)
\end{aligned}
$$

that we will compute explicitly below. The quantity we are interested in can be retrieved from the coefficient $z_1^M z_2^M$ of the power series expansion of the generating function,

$$F_M - G_M = Q_{M,M} = Q(z_1, z_2)|_{z_1^M z_2^M}, \quad (34)$$

where for simplicity here and below we use the same letter so that

$$G_N = G_{M(N)}, \quad F_N = F_{M(N)}, \quad (35)$$

where the relation between $N$ and $M$ was given above in (6). Note that while the $F_M, G_M$ are obtained here for any $M \geq 1$, the $F_N, G_N$ are obtained only for the specific values of $N$ corresponding to filled shells (the full dependence in $N \geq 1$ may be much more complex [31, 45] and is out of reach at present).

Note that $Q(z_1, z_2)$ contains the information both about the direct term and the exchange term, hence we will obtain $F_M$ and $G_M$ from a single calculation of $Q(z_1, z_2)$. The manipulations performed below on $Q$ will be such that the terms corresponding to $F_M$ and to $G_M$ will remain separate.

Let us introduce the center of mass and relative coordinate

$$\boldsymbol{a} = \frac{\boldsymbol{x} + \boldsymbol{y}}{2} \quad , \quad \boldsymbol{b} = \boldsymbol{x} - \boldsymbol{y}. \quad (36)$$

Inserting the expression (30) into (33) we obtain

$$
\begin{aligned}
Q(z_1, z_2) = {} & \frac{z_1}{1 - z_1} \frac{1}{\pi^{d/2}(1 - z_1^2)^{d/2}} \frac{z_2}{1 - z_2} \frac{1}{\pi^{d/2}(1 - z_2^2)^{d/2}} \\
& \times \int d\boldsymbol{a} d\boldsymbol{b} W(\boldsymbol{b}) \exp\left( -2 \frac{1 - z_1 z_2}{(1 + z_1)(1 + z_2)} \boldsymbol{a}^2 \right) \\
& \times \left[ \exp\left( -\frac{(1 - z_1 z_2) \boldsymbol{b}^2}{2(1 + z_1)(1 + z_2)} + \frac{2(z_1 - z_2) \boldsymbol{a} \cdot \boldsymbol{b}}{(1 + z_1)(1 + z_2)} \right) - \exp\left( -\frac{(1 - z_1 z_2) \boldsymbol{b}^2}{2(1 - z_1)(1 - z_2)} \right) \right]. \quad (37)
\end{aligned}
$$

Integrating over the position of the center of mass $\boldsymbol{a}$ we obtain a relatively simple and symmetric expression

$$
\begin{aligned}
Q(z_1, z_2) = {} & \frac{1}{(2\pi)^{d/2}} \frac{z_1 z_2}{(1 - z_1 z_2)^{d/2}} \frac{1}{((1 - z_1)(1 - z_2))^{1 + d/2}} \\
& \times \int d\boldsymbol{b} W(\boldsymbol{b}) \left[ \exp\left( -\frac{(1 - z_1)(1 - z_2)}{1 - z_1 z_2} \frac{\boldsymbol{b}^2}{2} \right) - \exp\left( -\frac{1 - z_1 z_2}{(1 - z_1)(1 - z_2)} \frac{\boldsymbol{b}^2}{2} \right) \right], \quad (38)
\end{aligned}
$$

where we used $\int d\mathbf{a}\, e^{-A\mathbf{a}^2+B\mathbf{a}\cdot\mathbf{b}} = (\frac{\pi}{A})^{d/2} e^{\mathbf{b}^2 B^2/(4A)}$. Note that the coefficient of the term $z_1 z_2$ in the expansion of $Q(z_1, z_2)$ vanishes since it corresponds to $N = M = M_1 = M_2 = 1$, i.e. to a single fermion with no interactions, $F_1 - G_1 = 0$.

Until now formula (38) is exact for any interaction potential $W(\mathbf{b})$ such that the integral converges. In (38) the first term corresponds to the direct term (leading to $F_M$) and the second to the exchange term (leading to $G_M$).

There are a number of interaction potentials for which (38) can be evaluated exactly. The simplest one is a Gaussian interaction, $W(\mathbf{b}) = \exp(-\frac{1}{2}\mathbf{b}^2)$, for which the integral is a simple Gaussian integral. Let us first write a formula for any potential of the form

$$W(\mathbf{b}) = \int_0^{+\infty} dt\, f(t) \exp\left(-\frac{t}{2}\mathbf{b}^2\right). \tag{39}$$

Note that $W(\mathbf{b})$ is the laplace transform of $f(t)$ with Laplace parameter $s = \mathbf{b}^2/2$. This contains the case of the power law potential $W(\mathbf{b}) = |\mathbf{b}|^{-n}$ on which we will focus below. Indeed one has for $n > 0$, from the known Laplace transform of a power-law function,

$$|\mathbf{b}|^{-n} = \int_0^{+\infty} \frac{dt}{t^{1-n/2} 2^{n/2} \Gamma\left(\frac{n}{2}\right)} e^{-\frac{t}{2}\mathbf{b}^2}. \tag{40}$$

Note that the case of long-range potentials with $n < 0$ can also be studied (e.g. for $-1 < n < 0$ replacing $e^{-\frac{t}{2}\mathbf{b}^2} \to e^{-\frac{t}{2}\mathbf{b}^2} - 1$ and so on), but will not be considered here. One obtains

$$Q(z_1, z_2) = \int_0^{+\infty} dt\, f(t) I(t, z_1, z_2), \tag{41}$$

$$I(t, z_1, z_2) = \frac{z_1 z_2}{(1 - z_1 z_2)^{d/2}} \frac{1}{((1-z_1)(1-z_2))^{1+d/2}} \tag{42}$$

$$\times \left[\left(t + \frac{(1-z_1)(1-z_2)}{1-z_1 z_2}\right)^{-d/2} - \left(t + \frac{1-z_1 z_2}{(1-z_1)(1-z_2)}\right)^{-d/2}\right].$$

Let us now specify to the power law potential $W(\mathbf{b}) = |\mathbf{b}|^{-n}$ which corresponds to $f(t) = \frac{t^{n/2-1}}{2^{n/2}\Gamma(\frac{n}{2})}$ and which leads to further simplification. For $d > n$ we can use the identity for $u > 0$ [41]

$$\frac{1}{\Gamma\left(\frac{n}{2}\right)} \int_0^{+\infty} \frac{dt}{t^{1-n/2}(t+u)^{d/2}} = \frac{\Gamma(\frac{d-n}{2})}{\Gamma(\frac{d}{2}) u^{\frac{d-n}{2}}}. \tag{43}$$

It can be extended for $d + 2 > n$ by analytic continuation, since we only need

$$\frac{1}{\Gamma\left(\frac{n}{2}\right)} \int_0^{+\infty} \frac{dt}{t^{1-n/2}} \left(\frac{1}{(t+u)^{d/2}} - \frac{1}{(t+1/u)^{d/2}}\right) = \frac{\Gamma(\frac{d-n}{2})}{\Gamma(\frac{d}{2})}\left(u^{\frac{n-d}{2}} - u^{\frac{d-n}{2}}\right), \tag{44}$$

with

$$u = \frac{(1-z_1)(1-z_2)}{1 - z_1 z_2}. \tag{45}$$

The l.h.s of (44) is a convergent integral for $d + 2 > n$ and the r.h.s. is analytic in the same domain, its value for $d = n$ being simply $-2(\ln u)/\Gamma(n/2)$. Using these identities we obtain the generating function for the power law interactions as

$$Q(z_1, z_2) = \frac{\Gamma(\frac{d-n}{2})}{2^{n/2}\Gamma(\frac{d}{2})} z_1 z_2 \Big( [(1-z_1)(1-z_2)]^{\frac{n}{2}-1-d} (1-z_1 z_2)^{-\frac{n}{2}}$$

$$- [(1-z_1)(1-z_2)]^{-\frac{n}{2}-1} (1-z_1 z_2)^{\frac{n}{2}-d} \Big), \tag{46}$$

valid jointly for $d + 2 > n$, and where the first term is the direct term and the second the exchange term, each being valid only for $d > n$. The simplicity of the result (46), consisting only of powers of $z_1 z_2$, $(1 - z_1)(1 - z_2)$ and $(1 - z_1 z_2)$, allows one to extract the coefficients and their asymptotics, see below.

Now we will extract the exact expressions for $F_M$ and $G_M$ from the relation (34), i.e. from the coefficient of $(z_1 z_2)^M$ in the power-series expansion of $Q(z_1, z_2)$. Consider the first term in (46). It can be decomposed into two factors from which we first separately extract the coefficient of $(z_1 z_2)^M$. For the first factor we use the identity

$$z(1 - z)^a = \sum_{k \geq 1} (-1)^{k+1} \binom{a}{k-1} z^k \,, \tag{47}$$

where

$$\binom{a}{k} = \frac{\Gamma(a+1)}{\Gamma(k+1)\Gamma(a-k+1)} \tag{48}$$

is the generalized binomial coefficient. Applying it with $a = -n/2$ and $z = z_1 z_2$ implies that

$$z_1 z_2 (1 - z_1 z_2)^{-\frac{n}{2}} \Big|_{(z_1 z_2)^M} = (-1)^{M+1} \binom{-n/2}{M-1} \,, \tag{49}$$

which gives the decomposition of the first factor. To deal with the second factor we use the identities

$$(1 - z)^b = \sum_{p \geq 0} (-1)^p \binom{b}{p} z^p \,, \tag{50}$$

$$[(1 - z_1)(1 - z_2)]^b \Big|_{\text{diag}} = \sum_{p \geq 0} \binom{b}{p}^2 (z_1 z_2)^p = {}_2F_1(-b, -b, 1, z_1 z_2) \,, \tag{51}$$

where the second line follows from the first (here $O|_{\text{diag}}$ means that we retain only the terms of the form $(z_1 z_2)^p$ in $O$), and ${}_2F_1(\cdots)$ denotes the hypergeometric function [41]. For $b = \frac{n}{2} - 1 - d$ it gives the coefficient of $(z_1 z_2)^M$ in the second factor. Putting now the two factors together we obtain, for $d > n$

$$F_M = \frac{\Gamma(\frac{d-n}{2})}{2^{n/2}\Gamma(\frac{d}{2})} \sum_{k=1}^{M} (-1)^{k+1} \binom{-n/2}{k-1} \binom{\frac{n}{2} - 1 - d}{M - k}^2 \,, \tag{52}$$

and

$$G_M = \frac{\Gamma(\frac{d-n}{2})}{2^{n/2}\Gamma(\frac{d}{2})} \sum_{k=1}^{M} (-1)^{k+1} \binom{\frac{n}{2} - d}{k-1} \binom{-\frac{n}{2} - 1}{M - k}^2 \,. \tag{53}$$

If considering the combination $F_M - G_M$ the formula can be extended to $d + 2 > n$ as discussed above.

We can also use the identity

$$\sum_{k=1}^{M} (-1)^{k+1} \binom{a}{k-1} \binom{b}{M-k}^2 = \binom{b}{M-1}^2 {}_3F_2(-a, 1 - M, 1 - M; b - M + 2, b - M + 2; 1), \tag{54}$$

where ${}_3F_2(\cdots)$ denotes the (generalized) hypergeometric function [41], to obtain closed ex-

pressions

$$
\begin{aligned}
F_M &= \frac{\Gamma(\frac{d-n}{2})}{2^{n/2}\Gamma(\frac{d}{2})}\binom{-d+\frac{n}{2}-1}{M-1}^2 \\
&\times \ _3F_2\left(1-M,1-M,\frac{n}{2};-d-M+\frac{n}{2}+1,-d-M+\frac{n}{2}+1;1\right),
\end{aligned}
\tag{55}
$$

$$
\begin{aligned}
G_M &= \frac{\Gamma(\frac{d-n}{2})}{2^{n/2}\Gamma(\frac{d}{2})}\binom{-\frac{n}{2}-1}{M-1}^2 \\
&\times \ _3F_2\left(1-M,1-M,d-\frac{n}{2};-M-\frac{n}{2}+1,-M-\frac{n}{2}+1;1\right).
\end{aligned}
\tag{56}
$$

An alternative representation of the result can be obtained by calculating the generating function

$$
Q(z) = \sum_{M\geq 1}(F_M - G_M)z^M,
\tag{57}
$$

which can be obtained directly from (46) and (51) since by definition, for any $a(y) = \sum_k a_k y^k$ and $b(y) = \sum_k b_k y^k$ one has $\sum_M [a(y)b(y)]|_{y^M} z^M = a(z)b(z)$. One obtains

$$
\begin{aligned}
Q(z) &= \frac{\Gamma\left(\frac{d-n}{2}\right)}{\Gamma\left(\frac{d}{2}\right)2^{n/2}}z\Big[\ _2F_1\left(d+1-\frac{n}{2},d+1-\frac{n}{2},1,z\right)(1-z)^{-n/2} \\
&\quad - \ _2F_1\left(1+\frac{n}{2},1+\frac{n}{2},1,z\right)(1-z)^{n/2-d}\Big],
\end{aligned}
\tag{58}
$$

where the first term gives the direct term and the second gives the exchange term.

## 3.2 $d=2$, $1/|x|$ interaction ($n=1$)

For $d=2$ and for $n=1$ (the $1/|x|$ interaction), Eqs. (52), (53) (55),(56) read

$$
\begin{aligned}
F_M &= \sqrt{\frac{\pi}{2}}\sum_{k=1}^{M}(-1)^{k+1}\binom{-\frac{1}{2}}{k-1}\binom{-\frac{5}{2}}{M-k}^2 \\
&= \sqrt{\frac{\pi}{2}}\binom{-\frac{5}{2}}{M-1}^2 \ _3F_2\left(1-M,1-M,\frac{1}{2};-M-\frac{1}{2},-M-\frac{1}{2};1\right),
\end{aligned}
\tag{59}
$$

$$
\begin{aligned}
G_M &= \sqrt{\frac{\pi}{2}}\sum_{k=1}^{M}(-1)^{k+1}\binom{-\frac{3}{2}}{k-1}\binom{-\frac{3}{2}}{M-k}^2 \\
&= \sqrt{\frac{\pi}{2}}\binom{-\frac{3}{2}}{M-1}^2 \ _3F_2\left(1-M,1-M,\frac{3}{2};-M+\frac{1}{2},-M+\frac{1}{2};1\right).
\end{aligned}
\tag{60}
$$

In order to analyze their behavior at $N \gg 1$ (or equivalently, $M \gg 1$), it is convenient to consider the generating function (58), which reads $Q(z) = Q_F(z) - Q_G(z)$ with

$$
Q_F(z) = \sum_{M\geq 1}F_M z^M = \sqrt{\frac{\pi}{2}}\ _2F_1\left(\frac{5}{2},\frac{5}{2},1,z\right)\frac{z}{(1-z)^{1/2}},
\tag{61}
$$

$$
Q_G(z) = \sum_{M\geq 1}G_M z^M = \sqrt{\frac{\pi}{2}}\ _2F_1\left(\frac{3}{2},\frac{3}{2},1,z\right)\frac{z}{(1-z)^{3/2}}.
\tag{62}
$$

We will now study their (divergent) behavior near $z=1$ and extract from it the large $M$ expansion of $F_M$ and $G_M$.

Let us start with $G_M$ to illustrate the method. We will use that

$$\sum_{M\geq 1} M^s z^M = \text{Li}_{-s}(z) \quad , \quad \sum_{M\geq 1} M^s \ln(M) z^M = \partial_s \text{Li}_{-s}(z), \tag{63}$$

together with the expansion of the polylogarithm function $\text{Li}_{-s}(z)$ near $z = 1$. The structure of this expansion is recalled in Appendix A. For $s > -1$ the leading behavior of the polylogarithm at $z = 1$ is divergent with

$$\text{Li}_{-s}(z) \simeq \Gamma(s+1)(1-z)^{-(s+1)}. \tag{64}$$

Let us denote $\eta = 1 - z$. The generating function (62) has the expansion for small $\eta > 0$

$$Q_G(z) = \frac{2\sqrt{\frac{2}{\pi}}}{\eta^{7/2}} - \frac{5}{\sqrt{2\pi}\,\eta^{5/2}} + \frac{-2\ln(\eta) + 11 + \ln(256)}{16\sqrt{2\pi}\,\eta^{3/2}} - \frac{-2\ln\eta - 5 + 8\ln 2}{64\sqrt{2\pi}\sqrt{\eta}} + O\left(\sqrt{\eta}\right). \tag{65}$$

Note that there is no constant, i.e. $O(\eta^0)$ term, and more generally there are only $\eta^{p+1/2}$ terms with integer $p$ in the series (with logarithmic components beginning at the third term). If these expansion coefficients can be reproduced from the expansion near $\eta = 0$ of the "trial" linear combination

$$\begin{aligned} Q_G^{\text{trial}}(z) &= a_{5/2}\text{Li}_{-5/2}(z) + a_{3/2}\text{Li}_{-3/2}(z) + (a_{1/2} + b_{1/2}\partial_s)\text{Li}_{-1/2}(z) \\ &+ (a_{-1/2} + b_{-1/2}\partial_s)\text{Li}_{1/2}(z) + O(\sqrt{\eta}), \end{aligned} \tag{66}$$

where the $a_j, b_j$ are to be determined, then one can conclude that

$$G_M = a_{5/2}M^{5/2} + a_{3/2}M^{3/2} + \left(a_{1/2} + b_{1/2}\ln M\right)M^{1/2} + \left(a_{-1/2} + b_{-1/2}\ln M\right)M^{-1/2} + o\left(M^{-1/2}\right). \tag{67}$$

Using Mathematica one finds that there is a unique set of coefficients which reproduces (65) up to and including the term $1/\sqrt{\eta}$, which leads to (for $d = 2$ and $n = 1$)

$$G_M = \frac{1}{\pi\sqrt{2}}\left[\frac{32}{15}M^{5/2} + \frac{8}{3}M^{3/2} + \frac{1}{4}(\ln M + c_2)M^{1/2} + \frac{1}{16}(\ln M + c_2')M^{-1/2}\right] + o(M^{-1/2}), \tag{68}$$

with $c_2 = 6\ln 2 + \gamma - \frac{13}{6}$ and $c_2' = 6\ln 2 + \gamma - \frac{17}{6}$.

**Remark.** One can push the procedure to higher order, e.g. introducing $a_{-3/2}$ and $b_{-3/2}$ terms. The next order correction is then found to be $\frac{1}{\pi\sqrt{2}}\frac{1}{2048}(\ln M + c_2'')M^{-3/2}$ with $c_2'' = 6\ln 2 + \gamma + \frac{281}{20}$. Note that the series for $Q_G^{\text{trial}}(z)$, when pushed to higher orders, also contain terms $\eta^p$ with positive integer powers, $p \geq 0$. Since there appear to be no such terms in the series for $Q_G(z)$, it implies that in addition to the correction terms of the form $M^{-p/2}(\ln M + c)$, $p > 1$, there are corrections to (68) which cancel such terms. These corrections, since they lead to analytic terms $\eta^p$ with positive integer, must decay faster than any power law in $1/M$.

Let us now write $G_M$ as a function of $N$. For the 2D HO, one has

$$N = \frac{M(M+1)}{2} \implies M = \frac{-1 + \sqrt{1+8N}}{2}.$$

Plugging this into (68), one obtains, as a function of the number of fermions $N$,

$$G_N \simeq \frac{1}{\pi\sqrt{2}}\left[\frac{64 \times 2^{1/4}}{15}N^{5/4} + \frac{N^{1/4}(\ln N + \gamma_2)}{4 \times 2^{3/4}} + \frac{N^{-3/4}\left(49\ln N + \gamma_2'\right)}{4096 \times 2^{3/4}}\right], \tag{69}$$

with $\gamma_2' = 637\ln 2 + 98\gamma - \frac{9877}{30}$. It is interesting to note that in this expansion, terms of order $N^{3/4}$ and $N^{-1/4}$ are absent (they cancel out).

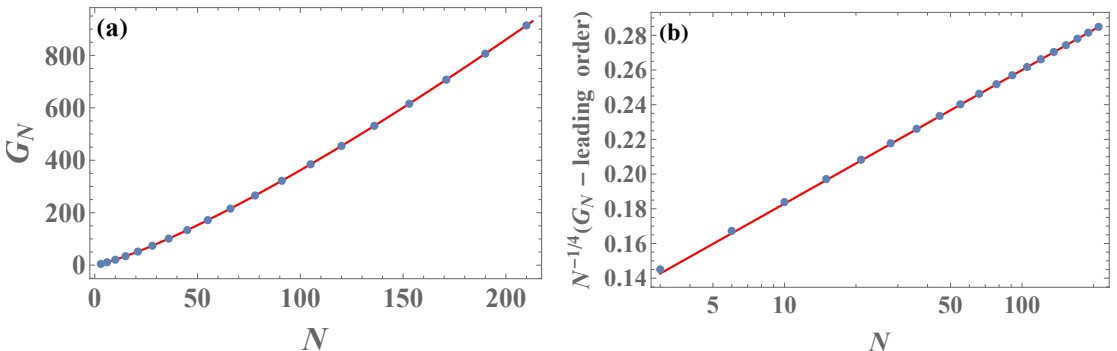

Figure 1: (a) $G_N$ vs. $N$ for the HO in $d = 2$. Markers are exact results (60) (where $N(M)$ is given by Eq. (6)). Solid line is the asymptotic behavior (26). (b) Markers are the exact result (60) minus the leading-order term ($\propto N^{5/4}$) in (26), rescaled by $N^{1/4}$, and solid line is the factor that multiplies $N^{1/4}$ in the remaining term in (26). Note the semi-log scaling in (b).

Similarly, let us now analyze the $N \gg 1$ behavior of the direct term. The expansion of $Q_F(z)$ near $z = 1$ has a very similar form

$$Q_F(z) = \frac{16\sqrt{\frac{2}{\pi}}}{3\eta^{9/2}} - \frac{28\sqrt{\frac{2}{\pi}}}{3\eta^{7/2}} + \frac{17}{2\sqrt{2\pi}\eta^{5/2}} - \frac{13}{24\sqrt{2\pi}\eta^{3/2}} + \frac{-36\ln(\eta) - 53 + 72\ln 4}{1536\sqrt{2\pi}\sqrt{\eta}} + O\left(\sqrt{\eta}\right). \tag{70}$$

Hence, we can use a similar trial form as in (66), except that we need to include a nonzero $a_{7/2}$ term. Applying the same method and using Mathematica we find

$$\begin{aligned}
F_M &= \frac{1}{\pi\sqrt{2}}\Bigg[\frac{512M^{7/2}}{315} + \frac{128M^{5/2}}{45} + \frac{10M^{3/2}}{9} - \frac{M^{1/2}}{18} \\
&+ \frac{3}{128}(\ln M + d_2)M^{-1/2} + O(M^{-3/2}\ln M)\Bigg],
\end{aligned} \tag{71}$$

with $d_2 = 6\ln 2 + \gamma - \frac{1451}{540}$. Expressed in terms of the number of fermions $N$ this leads to

$$F_N = \frac{1024 \times 2^{1/4}N^{7/4}}{315\pi} + \frac{2^{5/4}N^{3/4}}{45\pi} + \frac{3}{256\pi 2^{3/4}}(\ln N + \delta_2)N^{-1/4} + O(N^{-3/4}), \tag{72}$$

with $\delta_2 = 13\ln 2 + 2\gamma - \frac{133}{18}$ . The first few terms in these expansions are those which are reported in Eqs. (25) and (26) of the introduction. We compare the leading and subleading terms of $G_N$ to the exact result (60) in Fig. 1, showing excellent agreement. Indeed, the large-$N$ approximation works surprisingly well even for small $N$. For the case of a single (spinless) fermion, $N = 1$, the exact result (60) is $\sqrt{\pi/2} = 1.2533\ldots$, while the approximation (26) yields $1.2479\ldots$, i.e., it is accurate to within less than 0.5%. In fact, even just the leading order term in (26) gives a reasonable approximation, $32 \times 2^{3/4}/(15\pi) = 1.1420\ldots$, which is within 10% of the exact result. We also checked the next-order term in the expansion of $G_N$ [the last term in Eq. (69)], and performed analogous comparisons of $F_N$ with the asymptotic behaviors, also finding excellent agreement (not shown).

### 3.3   $d = 3$, $1/|x|$ interaction ($n = 1$)

Similarly, for $d = 3$ and for $n = 1$ the direct and exchange terms are given by

$$
\begin{aligned}
F_M &= \sqrt{\frac{2}{\pi}} \sum_{k=1}^{M} (-1)^{k+1} \binom{-1/2}{k-1} \binom{-\frac{7}{2}}{M-k}^2 \\
&= \sqrt{\frac{2}{\pi}} \binom{-\frac{7}{2}}{M-1}^2 {}_3F_2\left(1-M, 1-M, \frac{1}{2}; -M-\frac{3}{2}, -M-\frac{3}{2}; 1\right),
\end{aligned}
\tag{73}
$$

$$
\begin{aligned}
G_M &= \sqrt{\frac{2}{\pi}} \sum_{k=1}^{M} (-1)^{k+1} \binom{-\frac{5}{2}}{k-1} \binom{-\frac{3}{2}}{M-k}^2 \\
&= \sqrt{\frac{2}{\pi}} \binom{-\frac{3}{2}}{M-1}^2 {}_3F_2\left(1-M, 1-M, \frac{5}{2}; -M+\frac{1}{2}, -M+\frac{1}{2}; 1\right),
\end{aligned}
\tag{74}
$$

respectively, while the generating function (58) reads $Q(z) = Q_F(z) - Q_G(z)$ with

$$
Q_F(z) = \sum_{M \geq 1} F_M z^M = \sqrt{\frac{2}{\pi}} \, {}_2F_1\left(\frac{7}{2}, \frac{7}{2}; 1; z\right) \frac{z}{\sqrt{1-z}},
\tag{75}
$$

$$
Q_G(z) = \sum_{M \geq 1} G_M z^M = \sqrt{\frac{2}{\pi}} \, {}_2F_1\left(\frac{3}{2}, \frac{3}{2}; 1; z\right) \frac{z}{(1-z)^{5/2}}.
\tag{76}
$$

The same method as in the previous section leads to

$$
\begin{aligned}
G_M &= \frac{\sqrt{2}}{\pi^2} \Bigg[ \frac{64M^{7/2}}{105} + \frac{32M^{5/2}}{15} + \frac{M^{3/2}(\ln M + c_3)}{6} \\
&\quad + \frac{1}{4}M^{1/2}(\ln M + c_3') + \frac{125}{3072}M^{-1/2}(\ln M + c_3'') + O(M^{-3/2}\ln M) \Bigg],
\end{aligned}
\tag{77}
$$

with

$$
c_3 = 6\ln 2 + \gamma + \frac{47}{6} \quad , \quad c_3' = 6\ln 2 + \gamma - \frac{13}{6} \quad , \quad c_3'' = 6\ln 2 + \gamma - \frac{6851}{2500}.
\tag{78}
$$

One also obtains

$$
\begin{aligned}
F_M &= \frac{1}{\pi^2\sqrt{2}} \Bigg[ \frac{65536M^{11/2}}{155925} + \frac{32768M^{9/2}}{14175} + \frac{20864M^{7/2}}{4725} + \frac{448M^{5/2}}{135} \\
&\quad + \frac{1903M^{3/2}}{2700} - \frac{307}{5400}M^{1/2} + \frac{5}{512}(\ln M + d_3)M^{-1/2} + O(M^{-3/2}\ln M) \Bigg],
\end{aligned}
\tag{79}
$$

with $d_3 = 6\ln 2 + \gamma - \frac{549893}{283500}$.

Let us now express these results as a function of $N$. In $d = 3$ one has $N = \frac{M(M+1)(M+2)}{6}$ which leads to $M = \frac{\nu}{3^{2/3}} + \frac{1}{\nu 3^{1/3}} - 1$ with $\nu = \left(27N + \sqrt{3(243N^2 - 1)}\right)^{1/3}$. Substituting in the above expressions one finds

$$
\begin{aligned}
G_N &= \frac{2}{\pi^2} \Bigg[ \frac{64 \times 2^{2/3} 3^{1/6} N^{7/6}}{35} + \frac{N^{1/2}(\ln N + \gamma_3)}{6\sqrt{3}} \\
&\quad + \frac{7 \times 3^{5/6}}{1024 \times 2^{2/3}}(\ln N + \gamma_3')N^{-1/6} + O(N^{-1/2}\ln N) \Bigg],
\end{aligned}
\tag{80}
$$

with $\gamma_3 = \ln(3) + 19\ln(2) + 3\gamma - \frac{117}{10}$, and $\gamma_3' = \ln(3) + 19\ln 2 + 3\gamma - \frac{145499}{11340}$, and

$$
\begin{aligned}
F_N &= \frac{1}{\pi^2} \Bigg[ \frac{131072 \times 2^{1/3} N^{11/6}}{17325 \times 3^{1/6}} - \frac{128 \times 2^{2/3} N^{7/6}}{945 \times 3^{5/6}} + \frac{67N^{1/2}}{2100\sqrt{3}} \\
&\quad + \frac{5}{1536 \times 2^{2/3} \times 3^{1/6}}(\ln N + \delta_3)N^{-1/6} + O(N^{-1/2}\ln N) \Bigg],
\end{aligned}
\tag{81}
$$

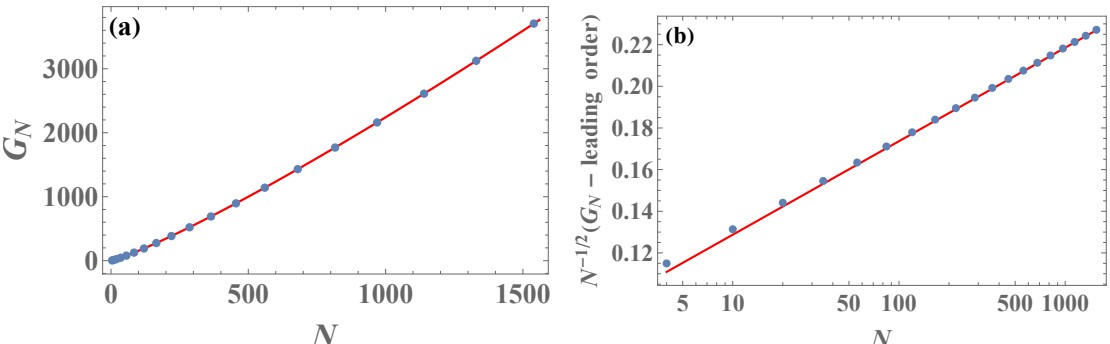

Figure 2: (a) $G_N$ vs. $N$ for the HO in $d = 3$. Markers are exact results (74) (where $N(M)$ is given by Eq. (6)). Solid line is the asymptotic behavior (28). (b) Markers are the exact result (74) minus the leading-order term ($\propto N^{7/6}$) in (28), rescaled by $N^{1/2}$, and solid line is the factor that multiplies $N^{1/2}$ in the remaining term in (28). Note the semi-log scaling.

with $\delta_3 = \ln(3) + 19\ln 2 + 3\gamma - \frac{96340693}{7654500}$. The first few terms in these expansions are those which are reported in Eqs. (27) and (28) of the introduction. We compare the leading and subleading terms of $G_N$ to the exact result (74) in Fig. 2, showing excellent agreement. Again we find that the large-$N$ approximations work very well even for small $N$: For $N = 1$, the exact result (74) is $\sqrt{2/\pi} = 0.79788\ldots$. In comparison, the leading-order term in (28) yields $\frac{128 \times 2^{2/3} 3^{1/6}}{35\pi^2} = 0.70639\ldots$ and the full expression (28) evaluates to $0.79024\ldots$, i.e., within 13% and 1% of the exact result, respectively. We also checked the next-order term in the expansion of $G_N$, including the $O(N^{-1/6})$ term in (80), and performed analogous comparisons of $F_N$ with the asymptotic behaviors, e.g. checking the first five terms in (79), with excellent agreement found here as well (not shown).

## 3.4 $\quad d = n = 1$

The general case $n = d$ is a little delicate from a technical point of view (the limit $n \to d$ in the general results for $n < d + 2$ must be taken carefully), and is treated in Appendix B. For $n = d = 1$, we obtain

$$E_N^{(1)} = \sum_{k=1}^{N} \frac{2\sqrt{2}\,\Gamma\left(k - \frac{1}{2}\right)\Gamma\left(-k + N + \frac{3}{2}\right)^2 \left(2\mathcal{H}_{-k+N+\frac{1}{2}} - \mathcal{H}_{k-\frac{3}{2}} - 4 + \ln 4\right)}{\pi^2 \Gamma(k)\Gamma(-k + N + 1)^2}, \qquad (82)$$

where the $\mathcal{H}_k$'s denote harmonic numbers, i.e., $\mathcal{H}_k = \psi(k+1) + \gamma$ where $\psi(z) = \frac{d}{dz}\ln\Gamma(z)$ is the digamma function. Extracting the large-$N$ behavior from Eq. (82) (see Appendix B for the details) we obtain Eq. (24) reported above. In the next section, we rederive the asymptotic behavior (24) using approximate methods.

# 4 Calculating $E_N^{(1)}$ for $n = 1$ and $N \gg 1$ via leading-order semiclassics

We would now like to gain some understanding of the physical origin of the terms in the large-$N$ behaviors reported above. Of particular interest is the exchange energy $G_N$ for $n = 1$ and $d = 2, 3$. The Dirac extension of the Thomas-Fermi model [46,47], gives the leading-order term

in $G_N$ from the semiclassical approximation (for completeness, we reproduce this result below; See also Ref. [48] for a proof that this is accurate for finite interactions). The first logarithmic correction to this result is not easily obtained from known corrections to the semiclassical approximation, and will be studied separately [40]. We also consider the case $n = d = 1$. Here, one cannot write $E_N^{(1)} = (F_N - G_N)/2$ as the difference between direct and exchange energies (because they each diverge), so the analysis is quite different. The approximations used in this section do not rely on the exact solvability of the model, and therefore may be useful for studying other models too (e.g., with anharmonic trapping potentials).

## 4.1 $d = 2$ and $d = 3$

It is easy to reproduce the leading-order large-$N$ behaviors of our results. In general $d$, in the absence of interactions, the semiclassical large-$N$ formula for the density is (e.g. [4,5])

$$N\rho_N(\boldsymbol{x}) \simeq \frac{(\mu - V(\boldsymbol{x}))_+^{d/2}}{(2\pi)^{d/2}\,\Gamma\left(1 + \frac{d}{2}\right)}. \tag{83}$$

Here and below we denote $(x)_+ = \max\{x, 0\}$. The spatial domain defined by $V(\boldsymbol{x}) < \mu$ is referred to as the bulk of the Fermi gas, while its boundary, given by $V(\boldsymbol{x}) = \mu$, is called the edge. At microscopic $|\boldsymbol{x} - \boldsymbol{y}|$ in the bulk, the kernel (11) takes the scaling form

$$K_N(\boldsymbol{x}, \boldsymbol{y}) \simeq \frac{1}{\ell(\boldsymbol{x})^d}\mathcal{K}_d^{\text{bulk}}\left(\frac{|\boldsymbol{x} - \boldsymbol{y}|}{\ell(\boldsymbol{x})}\right), \tag{84}$$

where

$$\ell(\boldsymbol{x}) = [N\rho_N(\boldsymbol{x})\gamma_d]^{-1/d}, \qquad \gamma_d = \frac{S_d}{d} = \pi^{d/2}\Gamma\left(\frac{d}{2} + 1\right). \tag{85}$$

The scaling function is

$$\mathcal{K}_d^{\text{bulk}}(x) = \frac{J_{d/2}(2x)}{(\pi x)^{d/2}}, \tag{86}$$

where $J_{d/2}$ is the Bessel function. At the origin, the scaling function takes the value $\mathcal{K}_d^{\text{bulk}}(0) = 1/\gamma_d$.

By plugging these approximations into Eqs. (22) and (23), one can calculate the leading-order large-$N$ behaviors of the direct and exchange terms $F_N$ and $G_N$, respectively. One finds that the exchange term is given, in $d = 2$, for general trapping potential, by (see Appendix C)

$$G_N \simeq \frac{16}{3\pi^2}\int \ell(\boldsymbol{x})^{-3}\,d\boldsymbol{x} = \frac{16}{3\pi^2}\int \left[\pi N\rho_N(\boldsymbol{x})\right]^{3/2}d\boldsymbol{x}. \tag{87}$$

Similarly, for $d = 3$ it is given by

$$G_N \simeq \frac{3^{4/3}}{4^{1/3}\pi^{1/3}}\int \left[N\rho_N(\boldsymbol{x})\right]^{4/3}d\boldsymbol{x}, \tag{88}$$

in agreement[4] with e.g., Eq. (4) in [36].

---

[4]The numerical coefficient $\frac{3^{4/3}}{4^{1/3}\pi^{1/3}}$ in our Eq. (88) differs from the corresponding coefficient in Eq. (4) in [36], which is $\frac{3^{4/3}}{4\pi^{1/3}}$, because in the present work we take the fermions to be spinless, whereas in [36] they have spin 1/2. This results in factor of 2 differences in the definitions of the density and of the exchange energy [recall also the factor 1/2 in (21)].

For the harmonic trapping potential, after plugging in the semiclassical density (83), the integrals (87) and (88) can be calculated (see Appendix C) and one obtains

$$
G_N \simeq \begin{cases} \frac{16\sqrt{2}}{15\pi}\mu^{5/2}, & d = 2, \\[2mm] \frac{64\sqrt{2}\mu^{7/2}}{105\pi^2}, & d = 3, \end{cases}
\tag{89}
$$

in perfect agreement with the leading-order terms in Eqs. (68) and (77), respectively, using that $\mu = M - 1 + d/2$, see (5). One can similarly calculate the direct term using semiclassical approximations , by plugging the density (83) into (22). This is a straightforward but technical calculation which we perform in Appendix D. The result is

$$
F_N \simeq \begin{cases} \frac{256\sqrt{2}}{315\pi}\mu^{7/2}, & d = 2, \\[2mm] \frac{32768\sqrt{2}}{155925\pi^2}\mu^{11/2}, & d = 3, \end{cases}
\tag{90}
$$

in perfect agreement with the leading-order terms in Eq. (71) and (79), respectively.

## 4.2   $d = 1$

In $d = 1$, we cannot separate the direct and exchange terms as we did for $d = 2, 3$. The semiclassical density (83) reads

$$
N\rho_N(x) \simeq \frac{\sqrt{2(\mu - V(x))_+}}{\pi} \simeq \frac{\sqrt{(2N - x^2)_+}}{\pi}.
\tag{91}
$$

At $x \simeq y$ in the bulk of the fermi gas, i.e., for $x - y \sim 1/\sqrt{N}$, that is of the order of the inter-particle distance in the bulk, the kernel is well approximated by the celebrated sine kernel

$$
K_N(x, y) \simeq \frac{\sin(k_F(x)|x - y|)}{\pi|x - y|},
\tag{92}
$$

where $k_F(x) = \sqrt{2(\mu - V(x))}$ is the local Fermi momentum.

We are now ready to evaluate the integral (20). First of all, since the integrand in (20) is invariant under exchanging $x$ and $y$, it is sufficient to integrate only over $y < x$ (and multiply the final result by 2). We partition the remaining integration domain into two subdomains: (i) $x \simeq y$, and (ii) $x$ and $y$ that are far from each other. Thus we write $E_N^{(1)} = I_1 + I_2$ where

$$
I_1 = \int_{-\infty}^{\infty} dx \int_{-\infty}^{x - \xi} dy \frac{N^2 \rho_N(x)\rho_N(y) - K_N(x, y)^2}{|x - y|},
\tag{93}
$$

$$
I_2 = \int_{-\infty}^{\infty} dx \int_{x - \xi}^{x} dy \frac{N^2 \rho_N(x)\rho_N(y) - K_N(x, y)^2}{|x - y|},
\tag{94}
$$

and $\xi$ is an intermediate "cutoff" $1/\sqrt{N} \ll \xi \ll \sqrt{N}$ (the result will not depend on the precise choice of $\xi$). Next, we calculate each of $I_1$ and $I_2$ using the approximations for the density and kernel given above (in $I_1$ it turns out that the term $K_N(x, y)^2$ is negligible in the leading order, see Appendix E for details):

$$
I_1 \simeq \frac{4\sqrt{2}N^{3/2}(-6\ln\xi + 3\ln N - 14 + 21\ln 2)}{9\pi^2},
\tag{95}
$$

$$
I_2 \simeq \frac{4\sqrt{2}N^{3/2}(6\ln\xi + 3\ln N + 6\gamma - 14 + 15\ln 2)}{9\pi^2}.
\tag{96}
$$

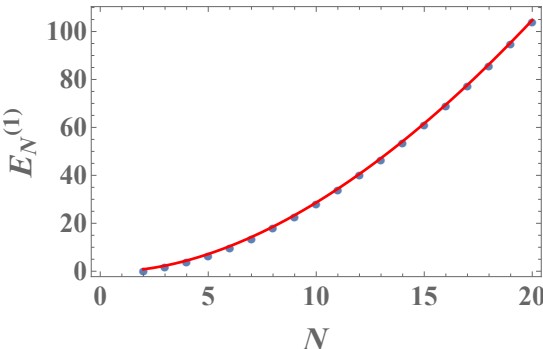

Figure 3: $E_N^{(1)}$ vs. $N$ for the HO in $d = 1$ and Coulomb interaction $n = 1$. Markers are exact result (82) and solid line is the large-$N$ asymptotic behavior (24).

Summing these two equations we obtain Eq. (24) reported above which, as we anticipated, does not depend on the choice of $\xi$. Eq. (24) is in good agreement with the exact result (82) at large $N$, see Fig. 3.

# 5   Perturbed density in the cases $d = 2, n = 1$ and $d = 3, n = 1$

Besides altering the ground-state energy, the interaction term in the Hamiltonian also affects other properties of the system, and in particular, the density. One approach for finding the modified density at $N \gg 1$ is to use semiclassical approximations with an effective potential that is given by the sum of the external potential and the effect of the interactions (for the case $d = 3, n = 1$ one recovers the usual Thomas-Fermi approximation). This is a fairly standard procedure. It was performed, e.g., in Refs. [49, 50] in the context of quantum dots in $d = 2$ for general trapping potentials and interaction strengths, for $n = 1$ [50] and for $n = 0^+$ (i.e., logarithmic interactions) [49]. Nevertheless, for completeness we present the results of this calculation here in our (relatively simple) setting, for the cases $d = 2, n = 1$ and $d = 3, n = 1$.

In the limit $N \gg 1$, this procedure gives an integral equation for the density (in presence of interactions) $N\rho_N$ and the effective potential $V_{\text{eff}}(x)$, through the two equations

$$N\rho_N(x) \simeq \frac{(\mu_{\text{eff}} - V_{\text{eff}}(x))^{d/2}}{(2\pi)^{d/2}\,\Gamma\left(1 + \frac{d}{2}\right)}, \tag{97}$$

$$V_{\text{eff}}(x) = V(x) + \epsilon \int N\rho_N(y)W(x,y)\,dy, \tag{98}$$

where $\mu_{\text{eff}}$ is found from the normalization $\int N\rho_N(x)\,dx = N$ of the density (so $\mu$ also changes due to the interaction).

Eqs. (97) and (98) are valid provided that the number of particles is large ($N \gg 1$) but they do not rely on the assumption that the interaction term is small. However, if we add the assumption $\epsilon \to 0$, the integral equation simplifies considerably. In Appendix F, we solve the Thomas-Fermi equations to first order in $\epsilon$. In $d = 2$, we obtain (in terms of the rescaled variable $X = x/\sqrt{2\mu}$)

$$N\rho_N\left(\sqrt{2\mu}\,X\right) \simeq \frac{\mu\left(1 - X^2\right) + \sqrt{2}\,\epsilon\mu^{3/2}\left(\frac{64}{45\pi} - v_1(X)\right)}{2\pi}, \tag{99}$$

with

$$v_1(X) = \frac{4(X-1)}{9\pi}\left[(X^2-2)E\left(-\frac{4X}{(X-1)^2}\right)-(X+1)^2 K\left(-\frac{4X}{(X-1)^2}\right)\right], \quad (100)$$

where $E(m)$ and $K(m)$ are the complete elliptic integrals of first and second kind, respectively [41]. In $d=3$, (and $n=1$), a special simplification occurs: The integral equation for the Coulomb potential can be transformed into the differential Poisson equation (more generally, this simplification occurs if $n=d-2$). Using this (see Appendix F), we obtain

$$N\rho_N(x) \simeq \frac{\sqrt{2}}{3\pi^2}\left[(\mu-V(x))^{3/2}+\frac{3}{2}\epsilon\sqrt{\mu-V(x)}\left(\frac{32768\times 2^{1/3}N^{5/6}}{4725\times 3^{1/6}\pi^2}-V_1(x)\right)\right], \quad (101)$$

where

$$V_1(x) = \frac{4\sqrt{2}}{3\pi}\left[\frac{\mu^3}{4\sqrt{2}x}\arctan\left(\frac{x}{\sqrt{2\mu-x^2}}\right)+\frac{1}{120}\sqrt{\mu-\frac{x^2}{2}}\left(33\mu^2+2x^4-13\mu x^2\right)\right]. \quad (102)$$

Incidentally, this calculation also enables us to obtain the leading-order behavior of $E_N^{(1)}$ at large $N$, from the relation $dE_N/dN \simeq \mu_{\text{eff}}$ which follows from the fact that $\mu_{\text{eff}}$ plays the role of an effective chemical potential. In Appendix F we calculate $\mu_{\text{eff}}$ in $d=2,3$ and show that it indeed coincides with the derivatives of the leading-order terms in Eqs. (25) and (27) with respect to $N$.

# 6 Simplifications in special cases

In this section we give some explicit results for the cases in which $n$ is even (and the space dimension $d$ is an integer). In these cases, the result (58) simplifies considerably, because the hypergeometric function $_2F_1(a,a,1,z)$ simplifies for integer $a$ into rational functions. Taking into account the constraint $n < d+2$, one finds the relevant cases in physical spatial dimension are $n=2$, $d \in \{1,2,3\}$, and $n=4$, $d=3$. The case $n=d=2$ is delicate, and treated in the Appendix B. We also point out at the end of this Section a remarkable symmetry which allows to obtain the result for $d=2, n=3$ with no further calculation. We now consider the various cases in detail.

## 6.1 The case $d=1, n=2$

The case $d=1$ and $n=2$ corresponds to the Calogero-Sutherland model [25,26]. In that case the many-body ground state is known exactly, and can be obtained for instance via an exact mapping to Gaussian random matrix ensembles. The ground-state energy reads [27]

$$E_N = \frac{\beta}{4}N(N-1)+\frac{N}{2}, \quad (103)$$

where $\epsilon = \beta(\beta-2)/4$. The noninteracting case, $\epsilon=0$, corresponds to $\beta=2$, and expanding the exact result in small $\epsilon$, one finds

$$E_N = \frac{1}{2}N^2+\frac{\epsilon}{2}N(N-1)+O\left(\epsilon^2\right). \quad (104)$$

The leading-order term is in agreement with Eqs. (6) and (7), and the $O(\epsilon)$ term is in agreement with our perturbative result, as we now show. Indeed, Eq. (58) reads, for $n=2, d=1$,

$$Q(z) = \frac{2z^2}{(1-z)^3} = \sum_{M=1}^{\infty} M(M-1)z^M, \quad (105)$$

from which one immediately extracts

$$E_M^{(1)} = \frac{M(M-1)}{2}, \tag{106}$$

which, using $N = M$, coincides with the $O(\epsilon)$ term in Eq. (104).

## 6.2 The cases $d = 3, n = 2$ and $d = 3, n = 4$

Let us consider $d = 3$ and $n = 2$. In this case, the condition $n < d$ holds, and thus one can separate the two terms in Eq. (58), to obtain

$$Q_F(z) = \frac{z(z^2 + 4z + 1)}{(1-z)^6} \quad , \quad Q_G(z) = \frac{z(z+1)}{(1-z)^5} \quad , \quad Q(z) = \frac{2z^2(z+2)}{(1-z)^6}, \tag{107}$$

which immediately leads to the polynomial forms

$$F_M = \frac{1}{60}M(M+1)(M+2)(3M^2 + 6M + 1), \tag{108}$$

$$G_M = \frac{1}{12}M(M+1)^2(M+2) \tag{109}$$

and

$$E_M^{(1)} = \frac{1}{2}(F_M - G_M) = \frac{1}{120}(M-1)M(M+1)(M+2)(3M+4). \tag{110}$$

Consider now $d = 3$ and $n = 4$. Amazingly, one again finds that Eq. (58) gives

$$Q(z) = \frac{2z^2(z+2)}{(1-z)^6}, \tag{111}$$

which is identical to the case $d = 3$, $n = 2$ (although now it is obtained via an analytical continuation which in that case is simple). This immediately implies that for $d = 3$ and $n = 4$ the energy correction $E_M^{(1)}$ is still given by the formula (110). Using the relation (6) between $N$ and $M$, we find that the large-$N$ expansion of the result (110) is

$$E_N^{(1)} = \frac{3^{5/3}N^{5/3}}{10 \times 2^{1/3}} - \frac{3^{1/3}N^{4/3}}{2^{5/3}} - \frac{N^{2/3}}{12 \times 6^{1/3}} + \frac{N^{1/3}}{60 \times 6^{2/3}} - \frac{N^{-1/3}}{1620 \times 6^{1/3}} + \dots . \tag{112}$$

Here, as we found also above for the case $n = 1$, it is interesting to note that it appears that there are terms "missing" from this expansion, namely the $O(N^1)$ and $O(N^0)$ terms.

In fact, more generally there is a similar surprising relation between the cases $(d, n_1)$ and $(d, n_2)$ if $d = (n_1 + n_2)/2$. Indeed, when evaluating Eq. (58) in these two cases one finds that the terms in the square brackets are simply exchanged, so one finds that the functions $Q(z)$ differ only by a multiplicative constant:

$$\frac{Q(z)|_{d,n_1}}{Q(z)|_{d,n_2}} = -2^{(n_2-n_1)/2} \frac{\Gamma\left(\frac{d-n_1}{2}\right)}{\Gamma\left(\frac{d-n_2}{2}\right)}. \tag{113}$$

As a result, the corresponding $E_N^{(1)}$'s for the two cases differ by the exact same multiplicative constant. In the particular case $d = 3, n_1 = 2, n_2 = 4$ considered above, this constant is unity.

### 6.3 The case $d = 2, n = 3$

Using Eq. (113) together with our results for $d = 2, n = 1$, we can immediately obtain the solution to the case $d = 2$ and $n = 3$. The constant of proportionality in Eq. (113) (for $d = 2, n_1 = 1, n_2 = 3$) again turns out to equal unity, and thus we obtain the rather remarkable result

$$E_N^{(1)}|_{d=2,n=3} = E_N^{(1)}|_{d=2,n=1}, \tag{114}$$

where, to remind the reader, $E_N^{(1)}|_{d=2,n=1}$ was calculated in subsection 3.2.

## 7 Discussion

To summarize, we studied a system of $N$ fermions trapped in a harmonic potential in general spatial dimension $d$, with power-law interactions which we assumed are weak ($\propto \epsilon$). Assuming that $N$ is such that the highest energy shell is full, we calculated the exact first-order correction $\epsilon E_N^{(1)}$ to the many-body ground state energy of the system. Wherever possible, we wrote $E_N^{(1)}$ as the difference between a direct and an exchange term, and calculated each of the two terms separately.

Focusing on the particular case of the Coulomb interaction $\propto 1/r$, we analyzed the $N \gg 1$ behavior of $E_N^{(1)}$, and found that, as expected, in $d > 1$ the leading order of the exchange term coincides with the result of the LDA — the Dirac expression for exchange — applied to the simple semiclassical-limit result for the density distribution. Interestingly, we found that the subleading correction to this term embodies a logarithmic divergence, as is known to be the case for electrons in atoms [36] (both neutral atoms and the Bohr atom). It would be useful to better understand the physical origin of each of the terms in the large-$N$ expansion of $E_N^{(1)}$ that we obtained here. In particular, the leading logarithmic correction to exchange is of direct relevance to DFT, and a separate study of it is forthcoming [40].

In this context of DFT, the efficiency of the large-$N$ expansion for exchange is noteworthy. Even for the smallest value of $N$ considered, a single full shell ($N = 1$), the leading term captures the exchange energy to within 13% for $d = 3$ using a single large-$N$ coefficient, and to within 1% using two additional large-$N$ coefficients, those of the first logarithmic correction and the corresponding power-of-$N$ term. (the corresponding results for $d = 2$ are 10% and 0.5%, respectively.

We also studied the leading-order effect of the interaction on the gas density at $N \gg 1$. It would be interesting to continue and extend our analysis by investigating the effect of the interactions on other properties of the interacting gas, such as the correlation energy, correlations of the density in real space and/or in momentum space, extreme-value statistics, counting statistics and entanglement entropy [14, 27, 52–55].

Several additional directions for future research remain. For instance, it would be interesting to extend our results to the case in which $N$ is such that the highest energy shell is only partly occupied, and degenerate perturbation theory becomes relevant. In this case, it is reasonable to expect additional correction terms with oscillations as a function of $N$ in analogy with atomic physics [31, 45].

It would be very interesting to extend our results to other trapping potentials, e.g., to atoms. While the exact method that we introduced may only be applied to special, exactly solvable cases, the approximate methods used here (especially in one spatial dimension) are expected to be more broadly applicable. Indeed, they may very well prove useful to extend our results to other cases (e.g., other trapping potentials and/or interactions) that do not have some underlying exactly solvable mathematical structure.

We gave explicit results for the case of power-law interactions $W(\boldsymbol{x},\boldsymbol{y}) = |\boldsymbol{x}-\boldsymbol{y}|^{-n}$. However, since $E_N^{(1)}$ is linear with respect to the interaction term $W(\boldsymbol{x},\boldsymbol{y})$, our results may be immediately extended to any interaction that can be written as the sum of such power laws. It would be interesting to extend our results even further, to more general types of interactions. The intermediate formula (41) that we obtained, which is valid for a large class of interactions, should provide a path in that direction, e.g. it allows to introduce a small scale cutoff.

One could try to extend our analysis to higher orders in the interaction strength $\epsilon$, or even try to go beyond the weakly-interacting regime. However, this appears to represent a significant challenge.

Finally, it is worth noting that, for $d = 1$, the noninteracting case can be exactly mapped to GUE random matrices (or equivalently, to a gas of classical particles at thermal equilibrium trapped by an external harmonic potential and interacting logarithmically) [19]. As a result, $E_N^{(1)}$ can be interpreted as the expectation value of the observable $\mathcal{W} = \sum_{1 \le i < j \le N} W(\lambda_i, \lambda_j)$ where $\lambda_1, \ldots, \lambda_N$ are the eigenvalues of a random GUE matrix. Such observables represent a natural extension to the "linear statistics" $\sum_{i=1}^{N} U(\lambda_i)$ that are often studied in random matrix theory and/or in the study of interacting classical gases [58,59]. In these contexts, as well as in the context of trapped fermions, it could be interesting to extend our results by studying the higher moments, and full distribution, of such observables $\mathcal{W}$.

# Acknowledgements

We thank G. Schehr for discussions related to Ref. [44].

**Funding information**   NRS acknowledges support from the Israel Science Foundation (ISF) through Grant No. 2651/23. PLD thanks the Ben Gurion university in the Negev for hospitality. PLD also thanks KITP for hospitality, supported in part by the National Science Foundation Grant No. NSF PHY-1748958 and PHY-2309135.

# A   Series expansion of polylogarithms

Let us recall here the structure of the expansions of the polylogarithm functions $\mathrm{Li}_{-s}(z)$ and their derivative with respect to $s$ near $z = 1$, as needed in the text.

With $z = 1 - x$ one has for $s > 0$ and non-integer, and $x > 0$ (see [60] combined with the expansion of $(-\ln(1-x))^{-s-1}$)

$$\sum_{n \ge 1} n^s z^n = \mathrm{Li}_{-s}(z) = A_s(x) + B_s(x), \tag{A.1}$$

$$A_s(x) = \Gamma(s+1) x^{-(s+1)} \left( 1 + \sum_{n \ge 1} c_{n,s} x^n \right), \tag{A.2}$$

$$c_{1,s} = \frac{1}{2}(-s-1), \quad c_{2,s} = \frac{1}{24}(s+1)(3s-2), \quad c_{3,s} = -\frac{1}{48}(s-2)(s-1)(s+1), \tag{A.3}$$

$$B_s(x) = \zeta(-s) + \sum_{n \ge 1} \frac{(-x)^n}{n!} \sum_{m=1}^{n} S_n^{(m)} \zeta(-s-m), \tag{A.4}$$

where $S_n^{(m)}$ are Stirling's number of the first kind. Taking a derivative w.r.t. $s$ one finds

$$\sum_{n\geq 1} n^s \ln(n) z^n = \partial_s \mathrm{Li}_{-s}(z) = \tilde{A}_s(x) + \tilde{B}_s(x),\tag{A.5}$$

$$\tilde{A}_s(x) = \Gamma(s+1)x^{-(s+1)}\bigg(\psi^{(0)}(s+1) - \ln x + \frac{1}{2}(s+1)x\left[\ln x - \psi^{(0)}(s+2)\right]$$
$$-\frac{1}{24}x^2(s+1)\left[(3s-1)\ln x - 3s\psi^{(0)}(s+2) + 2\psi^{(0)}(s+2) - 3\right] + O\left(x^3\right)\bigg),\tag{A.6}$$

$$\tilde{B}_s(x) = -\zeta'(-s) - \sum_{n\geq 1}\frac{(-x)^n}{n!}\sum_{m=1}^{n} S_n^{(m)}\zeta'(-s-m).\tag{A.7}$$

## B  General case $n = d$

### B.1  Exact result

In this Appendix we perform the analytical continuation to obtain the result for $n = d$. We give explicit formulae for $n = d = 1, 2, 3$.

Let us return to the formula (52) and (53) valid for $d > n$. Let us write the difference, expressing the binomial coefficients in terms of $\Gamma$ functions. One obtains

$$F_M - G_M = \frac{\Gamma(\frac{d-n}{2})}{2^{n/2}\Gamma(\frac{d}{2})}\sum_{k=1}^{M}(-1)^{k+1}\frac{A_{k,M}(d,n)}{\Gamma(k)\Gamma(-k+M+1)^2},\tag{B.1}$$

$$A_{k,M}(d,n) = \frac{\Gamma\left(1-\frac{n}{2}\right)\Gamma\left(\frac{n}{2}-d\right)^2}{\Gamma\left(-k-\frac{n}{2}+2\right)\Gamma\left(-d+k-M+\frac{n}{2}\right)^2}$$
$$-\frac{\Gamma\left(-\frac{n}{2}\right)^2\Gamma\left(-d+\frac{n}{2}+1\right)}{\Gamma\left(-d-k+\frac{n}{2}+2\right)\Gamma\left(k-M-\frac{n}{2}\right)^2}.\tag{B.2}$$

However this form is not suited to perform the limit $n = d$. Instead we transform all the $\Gamma$ functions using $\Gamma(x) = \pi/(\sin(\pi x)\Gamma(1-x))$, and simplify all the sine functions using explicitly that $k$ and $M$ are integers. This leads to

$$A_{k,M}(d,n) = (-1)^k\bigg(\frac{\Gamma\left(d+k-\frac{n}{2}-1\right)\Gamma\left(-k+M+\frac{n}{2}+1\right)^2}{\Gamma\left(\frac{n}{2}+1\right)^2\Gamma\left(d-\frac{n}{2}\right)}$$
$$-\frac{\Gamma\left(k+\frac{n}{2}-1\right)\Gamma\left(d-k+M-\frac{n}{2}+1\right)^2}{\Gamma\left(\frac{n}{2}\right)\Gamma\left(d-\frac{n}{2}+1\right)^2}\bigg).\tag{B.3}$$

Plugging Eq. (B.3) into (B.1), one can now take the limit $n = d$ and one finds

$$F_M - G_M = -\sum_{k=1}^{M}\frac{2^{-\frac{d}{2}-1}d\,\Gamma\left(\frac{d}{2}+k-1\right)\Gamma\left(\frac{d}{2}-k+M+1\right)^2}{\Gamma\left(\frac{d}{2}+1\right)^4\Gamma(k)\Gamma(-k+M+1)^2}$$
$$\times\left(d\left(-2\mathcal{H}_{\frac{d}{2}-k+M} + \mathcal{H}_{\frac{d}{2}+k-2} + \mathcal{H}_{\frac{d}{2}-1}\right) + 4\right),\tag{B.4}$$

where $\mathcal{H}_a = \psi(a+1) + \gamma$ is the Harmonic number and $\psi(x) = \frac{d}{dx}\ln\Gamma(x)$ the digamma function.

For $n = d = 1$ one finds

$$F_M - G_M = \sum_{k=1}^{M} \frac{4\sqrt{2}\,\Gamma\left(k - \frac{1}{2}\right)\Gamma\left(-k + M + \frac{3}{2}\right)^2 \left(2\mathcal{H}_{-k+M+\frac{1}{2}} - \mathcal{H}_{k-\frac{3}{2}} - 4 + \ln 4\right)}{\pi^2 \Gamma(k)\Gamma(-k+M+1)^2}, \qquad \text{(B.5)}$$

coinciding with Eq. (82) of the main text. For $n = d = 2$ one finds

$$F_M - G_M = \sum_{k=1}^{M}(-k + M + 1)^2 \left(2\mathcal{H}_{-k+M+1} - \mathcal{H}_{k-1} - 2\right), \qquad \text{(B.6)}$$

and for $n = d = 3$ one finds

$$F_M - G_M = \sum_{k=1}^{M} \frac{32\sqrt{2}\,\Gamma\left(k + \frac{1}{2}\right)\Gamma\left(-k + M + \frac{5}{2}\right)^2 \left(6\mathcal{H}_{-k+M+\frac{3}{2}} - 3\mathcal{H}_{k-\frac{1}{2}} - 10 + \ln 64\right)}{27\pi^2 \Gamma(k)\Gamma(-k+M+1)^2}. \qquad \text{(B.7)}$$

## B.2 Large-$N$ asymptotic behaviors

We start with Eq. (58), which we write here again for convenience:

$$\begin{aligned}
Q(z) &= \frac{\Gamma\left(\frac{d-n}{2}\right)}{\Gamma\left(\frac{d}{2}\right)2^{n/2}} z \Bigg[ {}_2F_1\left(d + 1 - \frac{n}{2}, d + 1 - \frac{n}{2}, 1, z\right)(1-z)^{-n/2} \\
&\quad - {}_2F_1\left(1 + \frac{n}{2}, 1 + \frac{n}{2}, 1, z\right)(1-z)^{n/2-d} \Bigg].
\end{aligned} \qquad \text{(B.8)}$$

Let us begin by analyzing the case $n = d = 1$. We first set $n = 1$. Then we write the expansion of $Q(z)$ in powers of $\eta = 1 - z$. It has the form

$$\begin{aligned}
Q(z) &= \eta^{-d-3/2}(a_0(d) + \eta a_1(d) + \dots) + \eta^{-2d-1/2}(b_0(d) + \eta b_1(d) + \dots) \\
&\quad + \eta^{-1/2}(c_0(d) + c_1(d)\eta + \dots) + e_0 + e_1\eta + \dots
\end{aligned} \qquad \text{(B.9)}$$

Each coefficient has poles at $d = 1$, however the first two series degenerate into each others, up to logarithms, in the limit $d \to 1$. Adding all terms of a given order in the $\eta$ expansion in that limit we find that all poles in $d - 1$ cancel and one obtains a finite limit, which reads

$$Q(z) = \frac{4\sqrt{2}(-\ln\eta - 2 + 4\ln 2)}{\pi^{3/2}\eta^{5/2}} - \frac{\sqrt{2}(-5\ln\eta - 8 + 20\ln 2)}{\pi^{3/2}\eta^{3/2}} + O\left(\frac{1}{\sqrt{\eta}}\right). \qquad \text{(B.10)}$$

Surprisingly we find that the first two terms can be reproduced by the series expansion of

$$Q(z) = b_{3/2}\partial_s \mathrm{Li}_{-s}(1-\eta)|_{s=3/2} + a_{3/2}\mathrm{Li}_{-3/2}(1-\eta) + O\left(\frac{1}{\sqrt{\eta}}\right), \qquad \text{(B.11)}$$

$$a_{3/2} = \frac{16\sqrt{2}\left(-\frac{14}{3} + \gamma + \ln 64\right)}{3\pi^2} \quad , \quad b_{3/2} = \frac{16\sqrt{2}}{3\pi^2}, \qquad \text{(B.12)}$$

which implies that for $d = n = 1$ one has

$$F_M - G_M = \left(a_{3/2} + b_{3/2}\ln M\right)M^{3/2} + O\left(M^{-1/2}, M^{-1/2}\ln M\right), \qquad \text{(B.13)}$$

i.e. the term $O(M^{1/2}, M^{1/2}\ln(M))$ vanishes. Note that the leading-order terms coincide with the result (24) reported above.

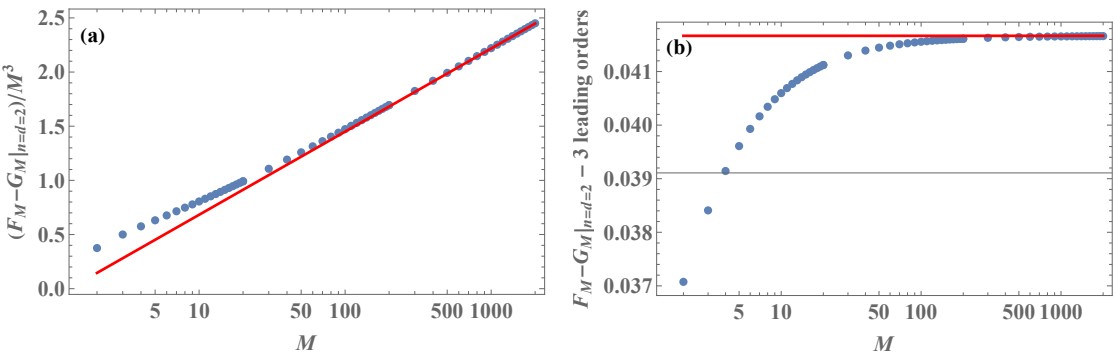

Figure 4: (a) $F_M - G_M$ vs. $M$ for $n = d = 2$. Markers represent the exact result (B.6), rescaled by $M^3$. Solid line is the corresponding leading-order asymptotic behavior $\frac{1}{3}\left(\ln M + \gamma - \frac{5}{6}\right)$ in Eq. (B.17). (b) Markers are the exact result (B.6) minus the three leading-order terms [up to and including the $O(M)$] in Eq. (B.17), and solid line is the next-order term, $1/24 = 0.4166\ldots$.

For $n = d = 2$ we first set $n = 2$. The we perform the expansion in $\eta = 1 - z$ for general $d$. Since it is a bit tricky to obtain we reproduce it here. One finds

$$-Q_G(z) = \frac{2}{2-d}\eta^{-2-d}\left(1 - \frac{3}{2}\eta + \frac{\eta^2}{2}\right), \tag{B.14}$$

$$Q_F(z) = -\frac{\pi\eta^{-2d}\csc(2\pi d)}{(d-2)\Gamma(2-2d)\Gamma(d)^2}\left(1 - \frac{d+1}{2}\eta\right.$$
$$\left. + \frac{(d-1)(d^2-2)\eta^2}{8d-12} + \frac{(d-2)(d-1)(d^2-3)\eta^3}{72-48d} + O(\eta^4)\right)$$
$$+ \frac{\pi\csc(2\pi d)}{(d-2)\eta\Gamma(1-d)^2\Gamma(2d)} + c_0 + c_1\eta + \ldots \tag{B.15}$$

[where $\csc(x) = 1/\sin(x)$]. Taking the limit $d \to 2$, all poles cancel and this simplifies into

$$Q(z) = -\frac{2(\ln\eta - 1)}{\eta^4} + \frac{3\ln\eta - 4}{\eta^3} + \frac{2 - \ln\eta}{\eta^2} + O(1). \tag{B.16}$$

Note that the term $O(1/\eta)$ cancels and that there is no $O(\ln\eta)$ term (the $O(\eta^4)$ term in the third line of (B.14) vanishes for $d = 2$). Using our standard method we finally obtain

$$F_M - G_M = \left(\ln M + \gamma - \frac{5}{6}\right)\frac{M^3}{3} + \left(\ln M + \gamma - \frac{1}{2}\right)\left(\frac{M^2}{2} + \frac{M}{6}\right) + \frac{1}{24} - \frac{\ln M}{90M} + O\left(\frac{1}{M}\right), \tag{B.17}$$

see Fig. 4. In terms of $N$ we obtain for $n = d = 2$

$$F_N - G_N = \frac{\sqrt{2}}{3}N^{3/2}(\ln N + \lambda_2) + \frac{N^{1/2}}{24\sqrt{2}}\left(\ln N + \lambda_2'\right) + \frac{1}{12} - \frac{79}{11520\sqrt{2}}\frac{\ln N}{\sqrt{N}} + O\left(\frac{1}{\sqrt{N}}\right), \tag{B.18}$$

with $\lambda_2 = \ln 2 + 2\gamma - \frac{5}{3}$ and $\lambda_2' = \ln 2 + 2\gamma - 7$.

Applying the same procedure for $n = d = 3$ we find

$$Q(z) = \frac{2\sqrt{2}}{\pi^{3/2}}\left(\frac{16(-3\ln\eta - 5 + 6\ln 4)}{9\eta^{11/2}} - \frac{4(-21\ln\eta - 29 + 42\ln 4)}{9\eta^{9/2}}\right. \tag{B.19}$$
$$+ \frac{-17\ln\eta - 14 + 34\ln 4}{4\eta^{7/2}} + \frac{13\ln\eta - 20 - 26\ln 4}{48\eta^{5/2}}$$
$$\left. + \frac{22\ln\eta - 127 - 44\ln 4}{6144\eta^{3/2}} + \frac{-690\ln\eta - 1951 + 1380\ln 4}{122880\sqrt{\eta}} + O(\sqrt{\eta})\right).$$

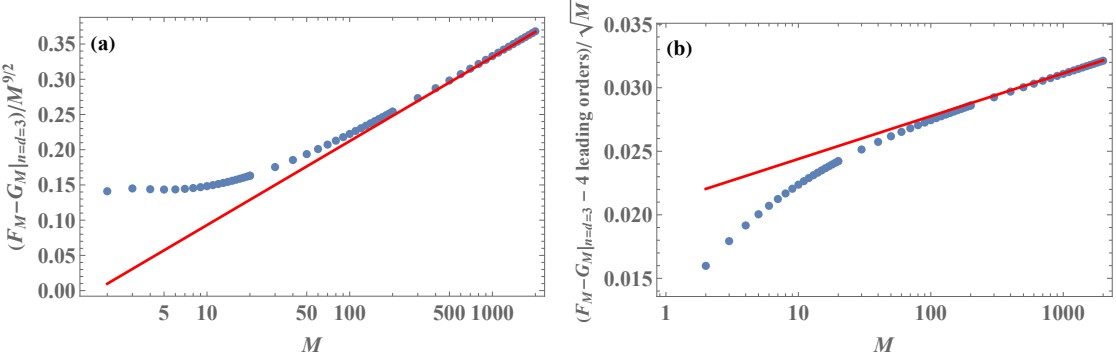

Figure 5: (a) $F_M - G_M$ vs. $M$ for $n = d = 3$. Markers represent the exact result (B.7), rescaled by $M^{9/2}$. Solid line is the corresponding leading-order asymptotic behavior $\frac{2\sqrt{2}}{\pi^2} \frac{512\left(\ln M + \gamma - \frac{1651}{315} + \ln 64\right)}{2835}$ in Eq. (B.20). (b) Markers are the exact result (B.7) minus the four leading-order terms [up to and including the $O(M^{3/2})$] in Eq. (B.20), rescaled by $\sqrt{M}$, and solid line is the corresponding next-order term, $\frac{2\sqrt{2}}{\pi^2} \frac{353\sqrt{M}\left(\ln M + \gamma + \frac{20395}{2118} + \ln 64\right)}{69120}$.

This leads to

$$
\begin{aligned}
F_M - G_M = \frac{2\sqrt{2}}{\pi^2} \Bigg( & \frac{512 M^{9/2}\left(\ln M + \gamma - \frac{1651}{315} + \ln 64\right)}{2835} \\
& + \frac{256}{315} M^{7/2}\left(\ln M + \gamma - \frac{527}{105} + \ln 64\right) + \frac{158}{135} M^{5/2}\left(\ln M + \gamma - \frac{5594}{1185} + \ln 64\right) \\
& + \frac{5}{9} M^{3/2}\left(\ln M + \gamma - \frac{64}{15} + \ln 64\right) + \frac{353\sqrt{M}\left(\ln M + \gamma + \frac{20395}{2118} + \ln 64\right)}{69120} \\
& - \frac{821\left(\ln M + \gamma - \frac{15037}{4926} + \ln 64\right)}{46080\sqrt{M}} + O(M^{-3/2}\ln M) \Bigg),
\end{aligned} \tag{B.20}
$$

see Fig. 5. In terms of $N$ we obtain for $n = d = 3$

$$
\begin{aligned}
F_N - G_N = \frac{2\sqrt{2}}{\pi^2} \Bigg[ & \frac{1024}{945}\sqrt{\frac{2}{3}} N^{3/2}(\ln N + \lambda_3) + \frac{2 \times 2^{5/6} N^{5/6}}{105 \times 3^{1/6}}(\ln N + \lambda_3') \\
& - \frac{437 N^{1/6}}{11520 \times 6^{5/6}}(\ln N + \lambda_3'') \Bigg] + O\left(\frac{\ln N}{N^{1/2}}\right)
\end{aligned} \tag{B.21}
$$

with $\lambda_3 = \ln 3 + 19\ln 2 + 3\gamma - \frac{1651}{105}$, $\lambda_3' = \ln 3 + 19\ln 2 + 3\gamma - \frac{842}{35}$, $\lambda_3'' = \ln 3 + 19\ln 2 + 3\gamma - \frac{64541}{4370}$.

## C Semiclassical calculation of the exchange term in $d = 2$ and $d = 3$

### C.1 General trapping potential

Plugging the bulk approximation (84) for the kernel into the definition (23) of the exchange term $G_N$, we obtain

$$
G_N = \iint \frac{K_N(\boldsymbol{x}, \boldsymbol{y})^2}{|\boldsymbol{x} - \boldsymbol{y}|} d\boldsymbol{x} d\boldsymbol{y} \simeq \iint \frac{1}{\ell(\boldsymbol{x})^{2d}} \mathcal{K}_d^{\text{bulk}} \left( \frac{|\boldsymbol{x} - \boldsymbol{y}|}{\ell(\boldsymbol{x})} \right)^2 \frac{1}{|\boldsymbol{x} - \boldsymbol{y}|} d\boldsymbol{x} d\boldsymbol{y} =
$$
$$
= \iint \frac{1}{\ell(\boldsymbol{x})^{2d}} \mathcal{K}_d^{\text{bulk}} \left( \frac{|\boldsymbol{x} - \boldsymbol{y}|}{\ell(\boldsymbol{x})} \right)^2 \frac{1}{\ell(\boldsymbol{x})} \frac{\ell(\boldsymbol{x})}{|\boldsymbol{x} - \boldsymbol{y}|} d\boldsymbol{x} \ell^d(\boldsymbol{x}) \frac{d\boldsymbol{y}}{\ell^d(\boldsymbol{x})} \underbrace{=}_{\boldsymbol{u} = \frac{\boldsymbol{y} - \boldsymbol{x}}{\ell(\boldsymbol{x})}}
$$
$$
= \int \frac{1}{\ell(\boldsymbol{x})^{d+1}} d\boldsymbol{x} \int \frac{\mathcal{K}_d^{\text{bulk}}(|\boldsymbol{u}|)^2}{|\boldsymbol{u}|} d\boldsymbol{u} \,. \tag{C.1}
$$

For $d = 2$, Eq. (C.1) becomes

$$
G_N \simeq \int \ell(\boldsymbol{x})^{-3} d\boldsymbol{x} \int_0^\infty \mathcal{K}_2^{\text{bulk}}(u)^2 \, 2\pi du \,. \tag{C.2}
$$

The $u$ integral can be calculated exactly,

$$
\int_0^\infty \mathcal{K}_2^{\text{bulk}}(u)^2 \, 2\pi du = 2\pi \int_0^\infty \left[ \frac{J_1(2u)}{\pi u} \right]^2 du = \frac{16}{3\pi^2} \,. \tag{C.3}
$$

So we get

$$
G_N \simeq \frac{16}{3\pi^2} \int \ell(\boldsymbol{x})^{-3} d\boldsymbol{x} \,. \tag{C.4}
$$

We can recast this in terms of the density, using

$$
\gamma_2 = \pi \Gamma(2) = \pi \implies \ell(\boldsymbol{x}) = [N \rho_N(\boldsymbol{x}) \gamma_2]^{-1/2} = [\pi N \rho_N(\boldsymbol{x})]^{-1/2} \,, \tag{C.5}
$$

so we get the general formula for $d = 2$

$$
G_N \simeq \frac{16}{3\pi^2} \int \ell(\boldsymbol{x})^{-3} d\boldsymbol{x} = \frac{16}{3\pi^2} \int [\pi N \rho_N(\boldsymbol{x})]^{3/2} d\boldsymbol{x} \,, \tag{C.6}
$$

coinciding with Eq. (87) of the main text.

In $d = 3$ Eq. (C.1) becomes

$$
G_N \simeq \int \ell(\boldsymbol{x})^{-4} d\boldsymbol{x} \int_0^\infty \mathcal{K}_3^{\text{bulk}}(u)^2 \, 4\pi u du \,. \tag{C.7}
$$

The integral over $u$ can be calculated exactly:

$$
\int_0^\infty \mathcal{K}_3^{\text{bulk}}(u)^2 \, 4\pi u du = 4 \int_0^\infty \frac{J_{3/2}(2u)^2}{(\pi u)^2} du = \frac{4}{\pi^3} \,. \tag{C.8}
$$

So we have the semiclassical result

$$
G_N \simeq \frac{4}{\pi^3} \int \ell(\boldsymbol{x})^{-4} d\boldsymbol{x} \,, \tag{C.9}
$$

which we can recast in terms of the density, using $\gamma_3 = \pi^{3/2} [\Gamma(3/2+1)] = \pi^{3/2} \frac{3\sqrt{\pi}}{4} = \frac{3\pi^2}{4}$, and

$$\ell(x) = [N\rho_N(x)\gamma_3]^{-1/3} \tag{C.10}$$

as

$$G_N \simeq \frac{4}{\pi^3} \int [N\rho_N(x)\gamma_3]^{4/3} dx = \frac{3^{4/3}}{4^{1/3}\pi^{1/3}} \int [N\rho_N(x)]^{4/3} dx, \tag{C.11}$$

which is Eq. (88) of the main text.

## C.2   Explicit results for the harmonic trapping potential

In $d=2$, for the harmonic oscillator $V(r) = r^2/2$, the semiclassical density (83) reads

$$N\rho_N(x) \simeq \frac{1}{2\pi} (\mu - V(x))_+ , \tag{C.12}$$

where $\mu$ is found from the normalization. The edge is at $r_{\text{edge}} = \sqrt{2\mu}$ so

$$N \simeq \int_0^{\sqrt{2\mu}} 2\pi r N\rho_N(r) dr = \frac{\mu^2}{2} \implies \mu \simeq (2N)^{1/2} . \tag{C.13}$$

And now, using the general formula (87) for $d=2$, we obtain

$$
\begin{aligned}
G_N &\simeq \frac{16}{3\pi^2} \int [\pi N\rho_N(x)]^{3/2} dx \simeq \frac{16}{3\pi^2} \int_0^{\sqrt{2\mu}} [\pi N\rho_N(r)]^{3/2} 2\pi r dr \\
&\simeq \frac{16}{3\pi^2} \int_0^{\sqrt{2\mu}} \left[ \pi \frac{1}{2\pi} \left( \mu - \frac{r^2}{2} \right) \right]^{3/2} 2\pi r dr = \frac{16\sqrt{2}}{15\pi} \mu^{5/2},
\end{aligned}
\tag{C.14}
$$

which is the first line of (89) of the main text.

In $d=3$, for the harmonic oscillator $V(r) = r^2/2$, the semiclassical density (83) reads

$$N\rho_N(r) = \frac{\sqrt{2}}{3\pi^2} \left( \mu - \frac{r^2}{2} \right)_+^{3/2} , \tag{C.15}$$

where $\mu$ is again found from the normalization. The edge is at $r_{\text{edge}} = \sqrt{2\mu}$ so

$$N = \int_0^{\sqrt{2\mu}} 4\pi r^2 N\rho_N(r) dr \simeq \frac{\mu^3}{6} \implies \mu \simeq (6N)^{1/3} . \tag{C.16}$$

And now, using the general formula (88) for $d=3$, we obtain

$$
\begin{aligned}
G_N &\simeq \frac{3^{4/3}}{4^{1/3}\pi^{1/3}} \int [N\rho_N(x)]^{4/3} dx \simeq \frac{3^{4/3}}{4^{1/3}\pi^{1/3}} \int_0^{\sqrt{2\mu}} 4\pi r^2 \left[ \frac{\sqrt{2}}{3\pi^2} \left( \mu - \frac{r^2}{2} \right)^{3/2} \right]^{4/3} dr \\
&= \frac{64\sqrt{2}\mu^{7/2}}{105\pi^2} ,
\end{aligned}
\tag{C.17}
$$

which is the second line of (89) of the main text.

# D  Semiclassical calculation of the direct term in $d = 2$ and $d = 3$

Here we calculate the direct term $F_N$, in the large-$N$ limit by using semiclassical approximations , for $d = 2$ and $d = 3$ and a Coulomb interaction $n = 1$.

## D.1  $d = 2$

Plugging (for $d = 2$) the semiclassical density (83) $N\rho_N(x) \simeq \frac{1}{2\pi}(\mu - V(x))$ into Eq. (22), we get

$$F_N = \iint \frac{N\rho_N(x)N\rho_N(y)}{|x-y|}dx\,dy \simeq \frac{1}{4\pi^2}\iint_{|x|,|y|\leq\sqrt{2\mu}} \frac{\left(\mu-\frac{x^2}{2}\right)\left(\mu-\frac{y^2}{2}\right)}{|x-y|}dx\,dy. \quad \text{(D.1)}$$

Changing the integration variables, $x = \sqrt{2\mu}X$, $y = \sqrt{2\mu}Y$, this becomes

$$\begin{aligned}
F_N &\simeq \frac{\mu^{7/2}}{\sqrt{2}\pi^2}\iint_{|X|,|Y|\leq 1} \frac{\left(1-X^2\right)\left(1-Y^2\right)}{|X-Y|}dX\,dY \\
&= \frac{\mu^{7/2}}{\sqrt{2}\pi^2}\iint_{|X|,|Y|\leq 1} \frac{\left(1-X^2\right)\left(1-Y^2\right)}{\sqrt{X^2+Y^2-2XY\cos\phi}}dX\,dY,
\end{aligned} \quad \text{(D.2)}$$

where $\phi$ is the angle between $X$ and $Y$, $X \cdot Y = XY\cos\phi$. Using polar coordinates, this integral becomes:

$$F_N \simeq \frac{\mu^{7/2}}{\sqrt{2}\pi^2}\int_0^1 2\pi X\,dX \int_0^1 Y\,dY \int_0^{2\pi} d\phi \frac{\left(1-X^2\right)\left(1-Y^2\right)}{\sqrt{X^2+Y^2-2XY\cos\phi}}, \quad \text{(D.3)}$$

where the factor of $2\pi$ comes from the integration over the polar angle of $X$. Changing the order of integration, we now perform the integrals over $X$ and $Y$ to obtain

$$\begin{aligned}
F_N \simeq &\frac{\sqrt{2}\mu^{7/2}}{\pi}\int_0^{2\pi} d\phi \frac{1}{210}\Bigg[30\cos(2\phi) + (34 - 20\cos\phi - 30\cos(2\phi))\sqrt{2-2\cos\phi} \\
&+ (15\cos(3\phi) - 47\cos\phi)\ln\left(\frac{\sqrt{1-\cos\phi}}{\sqrt{2}+\sqrt{1-\cos\phi}}\right) - 42\Bigg].
\end{aligned} \quad \text{(D.4)}$$

Integrating now over $\phi$, we obtain $F_N \simeq \frac{256\sqrt{2}}{315\pi}\mu^{7/2}$, which is the first line of (90) of the main text.

## D.2  $d = 3$

We rewrite the direct term (22) as

$$F_N \equiv \iint \frac{N\rho_N(x)N\rho_N(y)}{|x-y|}dx\,dy = \int dx\,N\rho_N(x)\mathcal{J}_N(x), \quad \mathcal{J}_N(x) = \int dy \frac{N\rho_N(y)}{|x-y|}. \quad \text{(D.5)}$$

Since $\rho_N(x) = \rho_N(x)$ is rotationally symmetric, so is $\mathcal{J}_N(x) = \mathcal{J}_N(x)$. We now use that in $d = 3$, $\nabla^2(1/|x|) = -4\pi\delta(x)$. Thus, applying the Laplace operator to $\mathcal{J}_N$ we obtain

$$\nabla^2 \mathcal{J}_N(x) = \frac{1}{x^2}\frac{d}{dx}\left(x^2\frac{d\mathcal{J}_N}{dx}\right) = -4\pi\int dy\,N\rho_N(y)\delta^3(x-y) = -4\pi N\rho_N(x). \quad \text{(D.6)}$$

We now use the semiclassical approximation for the density (83), which in $d = 3$ reads

$$N\rho_N(\boldsymbol{x}) \simeq \frac{\sqrt{2}}{3\pi^2}(\mu - V(\boldsymbol{x}))^{3/2}. \tag{D.7}$$

Plugging this into Eq. (D.6), we obtain

$$\frac{1}{x^2}\frac{d}{dx}\left(x^2\frac{d\mathcal{J}_N}{dx}\right) \simeq -\frac{4\sqrt{2}}{3\pi}\left(\mu - \frac{x^2}{2}\right)^{3/2}. \tag{D.8}$$

The solution to this differential equation is

$$\mathcal{J}_N(x) \simeq \frac{4\sqrt{2}}{3\pi}\left[\frac{\mu^3}{4\sqrt{2}x}\arctan\left(\frac{x}{\sqrt{2\mu - x^2}}\right) + \frac{1}{120}\sqrt{\mu - \frac{x^2}{2}}\left(33\mu^2 + 2x^4 - 13\mu x^2\right)\right], \tag{D.9}$$

where we determined the integration constants by requiring that

$$\mathcal{J}_N(0) = \int dy\,\frac{N\rho_N(y)}{|y|} = \int_0^{\sqrt{2\mu}} 4\pi y\,\frac{\sqrt{2}}{3\pi^2}\left(\mu - \frac{y^2}{2}\right)^{3/2}dy = \frac{8\sqrt{2}\mu^{5/2}}{15\pi}. \tag{D.10}$$

Plugging Eqs. (D.7) and (D.9) into the expression for $F_N$ in (D.5), we obtain

$$
\begin{aligned}
F_N &\simeq \int_0^{\sqrt{2\mu}} \frac{\sqrt{2}}{3\pi^2}\left(\mu - \frac{x^2}{2}\right)^{3/2}\frac{4\sqrt{2}}{3\pi} \\
&\quad \times \left[\frac{\mu^3}{4\sqrt{2}x}\arctan\left(\frac{x}{\sqrt{2\mu - x^2}}\right) + \frac{1}{120}\sqrt{\mu - \frac{x^2}{2}}\left(33\mu^2 + 2x^4 - 13\mu x^2\right)\right]4\pi x^2 dx \\
&= \int_0^1 \sqrt{2\mu}\,dX\,\frac{8\mu^5 X\left(X^2 - 1\right)}{135\pi^2} \\
&\quad \times \left[X\left(8X^6 - 34X^4 + 59X^2 + 48\sqrt{1-X^2} - 33\right) - 15\sqrt{1-X^2}\arctan\left(\frac{X}{\sqrt{1-X^2}}\right)\right] \\
&= \frac{32768\sqrt{2}}{155925\pi^2}\mu^{11/2}, \tag{D.11}
\end{aligned}
$$

which is the second line of (90) of the main text.

# E  Semiclassical calculation of $E_N^{(1)}$ for $d = n = 1$ (at $N \gg 1$)

At macroscopic $x - y$, one has $N^2\rho_N(x)\rho_N(y) \gg K_N(x,y)K_N(y,x)$. Neglecting the second term in the integral (93) we find that

$$I_1 \simeq \sqrt{2N}\int_{-1}^1 dX\int_{-1}^{X - \xi/\sqrt{2N}} dY\,\frac{2N\sqrt{(1-X^2)(1-Y^2)}}{\pi^2\,|X-Y|}, \tag{E.1}$$

where we have used the semiclassical density (91) and changed the integration variables $x = \sqrt{2N}X$, $y = \sqrt{2N}Y$. The integrals over $Y$ in (E.1) can be calculated exactly by using

$$
\begin{aligned}
\frac{2\sqrt{(1-X^2)(1-Y^2)}}{\pi^2\,|X-Y|} &= \frac{2i}{\pi}\partial_Y\left[i\sqrt{(1-X^2)(1-Y^2)} + X\sqrt{1-X^2}\ln\left(Y + i\sqrt{1-Y^2}\right)\right. \\
&\quad \left. + i\left(1-X^2\right)\ln\left(\frac{2i\left(\sqrt{(1-X^2)(1-Y^2)} + XY - 1\right)}{(1-X^2)^{3/2}(Y-X)}\right)\right]. \tag{E.2}
\end{aligned}
$$

The result is rather cumbersome so we will not present it here, but in the limit $\xi \ll \sqrt{2N}$ it simplifies considerably and we obtain

$$
\begin{aligned}
I_1 \simeq \frac{\sqrt{2}N^{3/2}}{\pi^2} \int_{-1}^{1} dX \Big[ &\left(1-X^2\right)\left(-2\ln\xi + \ln(8N) + 2\ln\left(1-X^2\right)-2\right) \\
&+ \sqrt{1-X^2}X\left(\pi + 2i\ln\left(X + i\sqrt{1-X^2}\right)\right)\Big].
\end{aligned} \tag{E.3}
$$

The integral over $X$ can now be performed exactly, leading to Eq. (95) of the main text.

We now calculate $I_2$. Inserting the semiclassical density (91) and the sine kernel (92) into the integral (94) and approximating $\rho_N(y) \simeq \rho_N(x)$ (which follows from $x \simeq y$), we obtain

$$
\begin{aligned}
I_2 &\simeq \int_{-\sqrt{2N}}^{\sqrt{2N}} dx \int_{x-\xi}^{x} dy \left[ \frac{2N-x^2}{\pi^2} - \frac{\sin^2\left(\sqrt{2N-x^2}(x-y)\right)}{\pi^2(x-y)^2} \right] \frac{1}{|x-y|} = \\
&= \int_{-\sqrt{2N}}^{\sqrt{2N}} dx \int_{-\xi}^{0} dz \left[ \frac{2N-x^2}{\pi^2} - \frac{\sin^2\left(\sqrt{2N-x^2}z\right)}{\pi^2 z^2} \right] \frac{1}{|z|}.
\end{aligned} \tag{E.4}
$$

The integral over $z$ can be solved exactly. Denoting $A = 2N - x^2$, we obtain

$$
\begin{aligned}
I_2 \simeq \frac{1}{4\pi^2\xi^2} \int_{-\sqrt{2N}}^{\sqrt{2N}} dx \Big[ &A\xi^2 \ln\left(16A^2\xi^4\right) - 4A\xi^2 \mathrm{Ci}\left(2\sqrt{A}\xi\right) + 2(2\gamma-3)A\xi^2 \\
&+ 2\sqrt{A}\xi\sin\left(2\sqrt{A}\xi\right) - \cos\left(2\sqrt{A}\xi\right) + 1 \Big],
\end{aligned} \tag{E.5}
$$

where $\mathrm{Ci}(z) = -\int_z^{\infty} \cos(t)/t\, dt$ is the cosine integral. Taking the leading-order asymptotic behavior of the integrand at $\xi \gg 1/\sqrt{A} \sim 1/\sqrt{N}$, this expression simplifies considerably, to

$$
I_2 \simeq \int_{-\sqrt{2N}}^{\sqrt{2N}} dx\, A \frac{2\ln\left(\xi\sqrt{A}\right) + 2\gamma - 3 + \ln(4)}{2\pi^2}. \tag{E.6}
$$

Finally, plugging back $A = 2N - x^2$ and performing the integral over $x$, we obtain Eq. (96) of the main text.

## F  Perturbed density

Here we will find the leading-order correction to the density due to the interactions in $d = 2, 3$ for $n = 1$ (Coulomb interactions). The starting point is the integral equation (97) and (98) of the main text. We now solve these equations perturbatively in $\epsilon$. At order 0 in $\epsilon$, one simply obtains $\mu_{\mathrm{eff}} = \mu$, $V_{\mathrm{eff}}(x) = V(x)$, and the density is $N\rho_N(x) \simeq \frac{1}{(2\pi)^{d/2}\Gamma\left(1+\frac{d}{2}\right)} (\mu - V(x))^{d/2}$. The leading-order $\epsilon > 0$ correction can be obtained by plugging this density into Eq. (98) which, for Coulomb interactions reads

$$
V_{\mathrm{eff}}(x) = V(x) + \epsilon \int \frac{N\rho_N(y)}{|x-y|} dy. \tag{F.1}
$$

From here onward, we treat the cases $d = 2$ and $d = 3$ separately.

## F.1  $d = 2$

For $d = 2$ the equation (F.1) reads (for the harmonic potential)

$$N\rho_N(\boldsymbol{x}) \simeq \frac{\mu_{\text{eff}} - V_{\text{eff}}(\boldsymbol{x})}{2\pi}, \qquad V_{\text{eff}}(\boldsymbol{x}) = \frac{x^2}{2} + \epsilon \int \frac{N\rho_N(\boldsymbol{y})}{|\boldsymbol{x} - \boldsymbol{y}|} d\boldsymbol{y}. \tag{F.2}$$

In the limit of small $\epsilon$, we obtain, by plugging in the unperturbed density $N\rho_N(\boldsymbol{x}) \simeq \frac{\mu - V(\boldsymbol{x})}{2\pi}$, the following perturbative expression for the effective potential:

$$V_{\text{eff}}\left(\sqrt{2\mu}\,\boldsymbol{X}\right) = \mu X^2 + \sqrt{2}\,\epsilon\mu^{3/2} v_1(\boldsymbol{X}), \quad v_1(\boldsymbol{X}) = \int \frac{1 - Y^2}{2\pi|\boldsymbol{X} - \boldsymbol{Y}|} d\boldsymbol{Y}. \tag{F.3}$$

The integral over $\boldsymbol{Y}$ can be solved exactly in polar coordinates, where $\phi$ is the angle between $\boldsymbol{X}$ and $\boldsymbol{Y}$

$$
\begin{aligned}
v_1(\boldsymbol{X}) &= \int_0^1 Y\,dY \int_0^{2\pi} d\phi \frac{1 - Y^2}{2\pi\sqrt{X^2 + Y^2 - 2XY\cos\phi}} = \\
&= -\int_0^{2\pi} \frac{d\phi}{24\pi}\Big\{\sqrt{X^2 - 2X\cos\phi + 1}\big[7X^2 + 5X(3X\cos(2\phi) + 2\cos(\phi)) - 8\big] \\
&\quad + 3X\cos\phi\big[5X^2\cos(2\phi) - X^2 - 4\big]\ln\left(\sqrt{X^2 - 2X\cos\phi + 1} - X\cos\phi + 1\right)\Big\} \\
\end{aligned}
\tag{F.4}
$$

$$= \frac{4(X - 1)\left[(X^2 - 2)E\left(-\frac{4X}{(X-1)^2}\right) - (X + 1)^2 K\left(-\frac{4X}{(X-1)^2}\right)\right]}{9\pi}, \tag{F.5}$$

where $E(m)$ and $K(m)$ are the complete elliptic integrals of first and second kind, respectively [41]. The density is therefore given by

$$N\rho_N\left(\sqrt{2\mu}\,\boldsymbol{X}\right) \simeq \frac{\mu_{\text{eff}} - \mu X^2 - \sqrt{2}\,\epsilon\mu^{3/2} v_1(X)}{2\pi}. \tag{F.6}$$

It remains to determine $\mu_{\text{eff}}$, which we do by requiring the normalization $N = \int_0^\infty 2\pi x N\rho_N(x)\,dx$ of the density. For this purpose, it is more convenient to use the expression (F.4) for $v_1(X)$. Then we can perform the integration first over $X$ and then over $\phi$ to obtain

$$
\begin{aligned}
\int_0^1 X v_1(X)\,dx &= \int_0^{2\pi} \frac{d\phi}{3600\pi}\Big\{60\sin\left(\frac{\phi}{2}\right)[17 - 15\cos(2\phi)] \\
&\quad + 45\cos(3\phi)\Big[1 - 5\ln\left(2\sin\left(\frac{\phi}{2}\right) - \cos\phi + 1\right)\Big] \\
&\quad + \cos\phi\Big[-600\sin\left(\frac{\phi}{2}\right) + 705\ln\left(2\sin\left(\frac{\phi}{2}\right) - \cos\phi + 1\right) - 173\Big] \\
&\quad - \frac{2[\cos\phi\ln(1 - \cos\phi) + 1]}{15}\Big\} = \frac{32}{45\pi}. \\
\end{aligned}
\tag{F.7}
$$

Using this we can now find the normalization of the density:

$$
\begin{aligned}
N &= 4\pi\mu \int_0^\infty X N\rho_N\left(\sqrt{2\mu}X\right) dX \simeq 2\mu \int_0^1 X\left[\mu_{\text{eff}} - \mu X^2 - \sqrt{2}\,\epsilon\mu^{3/2} v_1(X)\right] dX \\
&= 2\mu\left(\frac{\mu_{\text{eff}}}{2} - \frac{\mu}{4} - \frac{32\sqrt{2}\,\epsilon\mu^{3/2}}{45\pi}\right),
\end{aligned}
\tag{F.8}
$$

which leads to

$$\mu_{\text{eff}} \simeq \frac{N}{\mu} + \frac{\mu}{2} + \frac{64\sqrt{2}\,\epsilon\mu^{3/2}}{45\pi} \simeq \mu + \frac{64\sqrt{2}\,\epsilon\mu^{3/2}}{45\pi}\,. \tag{F.9}$$

Plugging this into Eq. (F.6), we obtain the density

$$N\rho_N\left(\sqrt{2\mu}\,X\right) \simeq \frac{\mu\left(1-X^2\right) + \sqrt{2}\,\epsilon\mu^{3/2}\left(\frac{64}{45\pi} - \nu_1(X)\right)}{2\pi}\,, \tag{F.10}$$

which is Eq. (99) in the main text.

As explained in the main text, this calculation also enables us to check our predictions for the ground-state energy at leading order in large $N$, via the relation $dE_N/dN \simeq \mu_{\text{eff}}$. From Eqs. (6) and (7) in $d = 2$, we have $E_N^{(0)} \simeq \frac{2\sqrt{2}N^{3/2}}{3}$ so $\frac{dE_N^{(0)}}{dN} \simeq \sqrt{2N}$, while from Eq. (25) we have

$$E_N^{(1)} \simeq \frac{F_N}{2} \simeq \frac{512 \times 2^{1/4}N^{7/4}}{315\pi} \implies \frac{dE_N^{(1)}}{dN} \simeq \frac{128 \times 2^{1/4}N^{3/4}}{45\pi}\,. \tag{F.11}$$

Indeed, by plugging $\mu \simeq \sqrt{2N}$ into Eq. (F.9) we find that $\mu_{\text{eff}} \simeq \frac{dE_N^{(0)}}{dN} + \epsilon\frac{dE_N^{(1)}}{dN} + O\left(\epsilon^2\right)$ as expected.

## F.2  $d = 3$

As noted in the main text, for $n = d - 2$ the integral equation for the effective potential can be transformed into a differential one. For $d = 3$ the Poisson equation may be obtained by taking the Laplacian of Eq. (F.1), one obtains

$$\nabla^2 V_{\text{eff}}(x) = \nabla^2 V(x) - 4\pi\epsilon N\rho_N(x)\,, \tag{F.12}$$

where we used that $\nabla^2\left(1/|x|\right) = -4\pi\delta(x)$. For the harmonic oscillator, $V(x) = x^2/2$ we get, by plugging in the unperturbed ($\epsilon = 0$) semiclassical density:

$$\frac{1}{x^2}\frac{d}{dx}\left(x^2\frac{d\left(V_{\text{eff}} - V\right)}{dx}\right) \simeq -\epsilon\frac{4\sqrt{2}}{3\pi}\left(\mu - \frac{x^2}{2}\right)^{3/2}\,. \tag{F.13}$$

By integrating this equation we obtain the leading-order correction to the effective potential,

$$V_{\text{eff}}(x) \simeq \frac{x^2}{2} + \epsilon V_1(x)\,, \tag{F.14}$$

$$V_1(x) = \frac{4\sqrt{2}}{3\pi}\left[\frac{\mu^3}{4\sqrt{2}x}\arctan\left(\frac{x}{\sqrt{2\mu - x^2}}\right) + \frac{1}{120}\sqrt{\mu - \frac{x^2}{2}}\left(33\mu^2 + 2x^4 - 13\mu x^2\right)\right]\,, \tag{F.15}$$

where we determined the integration constants by requiring

$$V_{\text{eff}}(0) = V(0) + \epsilon\int\frac{N\rho_N(y)}{|y|}dy \simeq \epsilon\int_0^{\sqrt{2\mu}}4\pi y\frac{\sqrt{2}}{3\pi^2}\left(\mu - \frac{y^2}{2}\right)^{3/2}dy = \epsilon\frac{8\sqrt{2}\mu^{5/2}}{15\pi} \tag{F.16}$$

(which is Eq. (F.1) with $x = 0$). From the effective potential, we obtain the density:

$$N\rho_N(x) \simeq \frac{\sqrt{2}}{3\pi^2}\left(\mu_{\text{eff}} - V_{\text{eff}}(x)\right)^{3/2} \simeq \frac{\sqrt{2}}{3\pi^2}\left[\left(\mu_{\text{eff}} - V(x)\right)^{3/2} - \frac{3}{2}\epsilon\sqrt{\mu_{\text{eff}} - V(x)}\,V_1(x)\right]\,. \tag{F.17}$$

We can now calculate $\mu_{\text{eff}}$ from the normalization of the density (F.17)

$$N \simeq \int_0^{\sqrt{2\mu}} 4\pi x^2 N \rho_N(x)\,dx \simeq \frac{\mu_{\text{eff}}^3}{6} - \epsilon \frac{8192\sqrt{2}\mu^{9/2}}{14175\pi^2}\,. \tag{F.18}$$

which we invert to get

$$\mu_{\text{eff}} \simeq (6N)^{1/3} + \epsilon \frac{32768 \times 2^{1/3}}{4725 \times 3^{1/6}\pi^2}N^{5/6}\,. \tag{F.19}$$

Finally, plugging this back into Eq. (F.17), we obtain

$$N\rho_N(x) \simeq \frac{\sqrt{2}}{3\pi^2}\left[(\mu - V(x))^{3/2} + \frac{3}{2}\epsilon\sqrt{\mu - V(x)}\left(\frac{32768 \times 2^{1/3}}{4725 \times 3^{1/6}\pi^2}N^{5/6} - V_1(x)\right)\right], \quad \text{(F.20)}$$

which is Eq. (101) of the main text. To remind the reader, $\mu \simeq (6N)^{1/3}$ is the fermi energy at $\epsilon = 0$.

Let us check that the relation $dE_N/dN \simeq \mu_{\text{eff}}$ holds. From Eqs. (6) and (7) in $d = 3$, we have

$$E_N^{(0)} \simeq \frac{3^{4/3}N^{4/3}}{2^{8/3}} \implies \frac{dE_N^{(0)}}{dN} \simeq \frac{3^{1/3}N^{1/3}}{2^{2/3}}, \tag{F.21}$$

and from Eq. (27) we have

$$E_N^{(1)} \simeq \frac{65536 \times 2^{1/3}N^{11/6}}{17325 \times 3^{1/6}\pi^2} \implies \frac{dE_N^{(1)}}{dN} \simeq \frac{32768 \times 2^{1/3}N^{5/6}}{4725 \times 3^{1/6}\pi^2}\,. \tag{F.22}$$

Indeed, by comparing with Eq. (F.19), $\mu_{\text{eff}} \simeq \frac{dE_N^{(0)}}{dN} + \epsilon \frac{dE_N^{(1)}}{dN} + O\left(\epsilon^2\right)$ as expected.

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
