# Peer review of "Exact first-order effect of interactions on the ground-state energy of harmonically-confined fermions"

_SciPost Physics, doi:SciPost Phys. 17, 038 (2024)_

## Round 2 · Referee Report · Anonymous (Referee 1) · 2024-6-7

Report on the revised version of "Exact first-order effect of interactions on the ground-state energy of harmonically-confined fermions"

I am happy with the revision and recommend to publish the present version.

Remaining typo after eq. (39): laplace (in capital please)

---

## Round 2 · Referee Report · Anonymous (Referee 2) · 2024-7-11

Strengths

I share the positive opinion of the previous referee, both on the originality and importance of the work and the numerous results contained in the present paper and also on the quality of the writing of the paper. The authors have in my opinion very satisfactorily addressed in their answer the points raised by the referee. The new version of the paper is thus suitable for publication.

Weaknesses

No weakness.

Report

I do recommend publication of the present version in SciPost Physics for the reasons mentioned in the other parts of this report.

Recommendation

Publish (surpasses expectations and criteria for this Journal; among top 10%)

---

## Round 2 · Author Response

Dear Editor,

Please find enclosed the revised version of our manuscript titled

"Exact first-order effect of interactions on the ground-state energy of harmonically-confined fermions"

by P. Le Doussal, N. R. Smith and N. Argaman,

that we would like to resubmit to SciPost.

Our paper was reviewed by one referee who recommended the publication in SciPost, after his/her comments have been suitably addressed. We have performed changes in accordance with the suggestions made by the referee. For convenience, a version of the manuscript in which the changes are marked in blue can be found in the following link:
https://drive.google.com/file/d/17niioEefwFTSnP3NJ0ESvxsQa2SMRWdl/view?usp=sharing

Below you will find a detailed reply to the referee's comments.

Sincerely yours,

P. Le Doussal, N. R. Smith and N. Argaman.

---

## Round 2 · List of Changes

Reply to comments by the referee

The authors perturbatively compute the first order correction E_N^{(1)} to the ground state energy E_N^{(0)} in certain many body quantum systems. The setup consists of N spinless Fermions in d=1,2 and 3 dimensions, confined by a harmonic potential, under the condition that all single energy levels are filled up to N. This provides a relation between the harmonic oscillator Fermi energy parametrized by an integer M and N, Eq. (5). The weak interaction is assumed to be translational invariant and of power-law type, sim epsilon/r^n, under the condition 0<n<d+2 for convergence. In the case of the Coulomb interaction in three dimensions with n=1, the asymptotic expansion of E_N^{(1)} is worked out up to several orders in fractional powers of N, including logarithmic corrections. It is based on the exact results for finite N, using generating functions. The leading and next-to-leading order are compared with numerical simulations, and the leading order is rederived from a semiclassical expansion in section 4. Several appendices with technical details round off the paper with an extensive introduction and discussion section. The paper is very well written and contains an impressive amount of detailed results, some of which hinting beyond the presented scheme. I have not been able to check all results in detail for correctness, in particular all the appendices, but the results seem to me sound and well founded, also in comparison to the numerics. I am not an expert in the detailed literature beyond leading order, but to me the results seem novel and certainly justify publication, also in comparison to the (re)derivation of existing results from semi-classics. Below follows a longer list of small corrections, typos and optional questions. It will not be difficult for the authors to take them into account, after which I strongly recommend publication in SciPost.

------> We thank the referee for recognizing the scientific merit of our work and summarizing the main points, and also for his/her very helpful comments, which we address below. For convenience, a version of the manuscript in which the changes are marked in blue can be found in the following link: https://drive.google.com/file/d/17niioEefwFTSnP3NJ0ESvxsQa2SMRWdl/view?usp=sharing

List of comments: 1. Whenever the authors speak of Coulomb interaction (n=1), they should add “three dimensional” Coulomb interaction, starting in line 6 of the abstract. This is because in one respectively two dimensions it is linear respectively logarithmic, as you may add somewhere.

------> We have corrected this point in the abstract, and also added a brief explanation shortly before explaining why we refer to the n=1 interaction as "Coulomb" throughout the paper (because it is indeed the electrostatic interaction in d=3 or in lower-dimensional systems embedded in a 3D space).

2. An extensive account is given for the expansion of the ground state energy of neutral atoms in the atomic number Z. From Section 2 onwards the role of the expansion parameter is played by N. However, no comment is made relating the two, please do.

------> We have added a short paragraph explaining this point towards the end of Sec. 1.2.

3. Page 2 line 7: its mean density is not spatially uniform. Add ”in general”, see e.g. [20] where it is.

------> Done

  1. End of section 1.1: perhaps add “temperature” to the list, see e.g. [19,23].

------> Done

5. It should be mentioned that the power law interaction in (4) is also called Riesz interaction, for d>n, also after Eq. (23). For the Riesz gas there is quite some recent mathematical literature, see e.g. [A] and references therein.

------> To our knowledge, the term "Riesz gas" usually refers to a gas of classical particles at nonzero temperature with power-law interactions. In this paper, we treat a gas of quantum particles at zero temperature, and therefore believe that it would be misleading to call it a Riesz gas.

6. Typos: in Eq. (9) the exponent should read – {bf x}^2/2 (or each factor should be pulled under the product).

------> Done

Likewise, the two terms x^2+y^2 in the exponent of Eq. (30) should be in bold

------> We use the convention that x denotes the modulus of the vector {bf x}. Thus, x^2 and {bf x}^2 are the same. We have added a comment after Eq. (4) introducing this notation.

Why are absolute values and complex conjugates introduced in Eqs. (10) and (11)? Although true in general, it seems to me that for the harmonic oscillator (HO) treated here, all wave functions are real, cf. (9). Therefore, the kernel in (11) and (31) will be automatically symmetric. Please comments on this.

------> The referee is correct, we have added a footnote to comment on this point, which we refer to just before Eq. (10). However, we did not change Eqs. (10) and (11), because in the general case the absolute value and complex conjugates may be important (in fact, this may even be true for the harmonic oscillator in d>1 if one chooses a different basis of eigenfunctions than the choice (9)).

7. It might be useful for the reader to state that that Eq. (39) is nothing but the Laplace transform, with s={bf b}^2/2, also in view of the comment about long range potentials, please add some details here.

------> We have added some suitable text between Eqs. (39) and (40).

8. I would generally give a reference to standard tables of integrals when it comes to identities such as Eq. (43), especially in view of the analytic continuation. This would also apply to standard sums in (6) or (7). Although one can check these e.g. with the software Mathematica, I am not satisfied with such a reference in general.

------> For the identities (6) and (43), We have added references to Gradsteyn's book. We obtained the identity (7) and were not able to find it in standard books, but, as we have added in a footnote, it is easy to prove it by induction on M.

9. After Eq. (59) (or even (51)) it might be useful to state the definition of these hypergeometric functions, also to see why 3F2 truncates I this case

------> We have added references to Gradshteyn's book in which the hypergeometric functions are defined, and its various simplified forms in cases where some of its arguments are integers are discussed. We preferred not to define the hypergeometric functions ourselves, precisely because one must define them carefully when some of their arguments are integers and this would be cumbersome.

10. At first I did not understand the logic of the asymptotic expansion between eqs. (61) and (68). Please try to explain this better as it is of crucial importance for the paper. As far as I understand there are two claims made here: I) An Ansatz can be made, such that the asymptotic expansion in eta = 1-z of (65) and (66) agree to a certain order (this will lead to a linear set of equations for the coefficients in (66), hence to the claim for the uniqueness – this cannot be proven with Mathematica but by linear algebra statements). II) The expansion in (67) can be read off by (asymptotically0 identifying M^s in Li_{-s}(z) from (63) Question: is this related to a well-known mathematical procedure, like Pade approximation, where the error could be made explicit? Notation: why is the notation in the error term in (67) change in (68)? Both should read o(M^{-1/2} because of potential logarithms, and not O(M^{-3/2}), correct?

------> It is an interesting point. We found this rather simple method but we are not aware of a well known mathematical procedure. As we discuss below (68) the leading error is obtained empirically by going to the next order. As we note there, we expect that there are even smaller additional error terms (smaller than any power of 1/M) but we do not know how to estimate them.

Concerning the notations we have made in Eq. (68) the change O(M^{-3/2}) -> o(M^{-1/2}) indeed.

11. Page 21 line 2 after (100): reference [48] does not provide a definition of the complete elliptic integrals E and K, please use NIST Handbook or Gradshteyn Rhizknik for example.

------> We have replaced the reference to the Wolfram website by a reference to Gradshteyn (it is in fact possible to find the definitions of the complete elliptic integrals on the Wolfram website but it is not very convenient and the definitions in Gradshteyn are presented better).

  1. Do you have an intuition why n=1 and n=3 agree to this order in (114)?

------> No, we do not have any intuition for this remarkable coincidence, which is a particular case of the more general Eq. (113). We believe that the text already conveys our surprise at Eqs. (111)-(114) sufficiently, so we did not make any changes here.

13. Appendix A: [55] does not quite provide the expansion (116), it has to be combined with the expansion of (-log(1-x))^{-s-1}, please give some more details. In (118) a factor 1/n! seems to be missing in the first sum.

------> We have indicated that indeed it should be combined with the expansion of the logarithm. We thank the Referee for pointed out a missing 1/n! in actually both (118) and (121) which we have rechecked and corrected.

14. Appendix B.2: in (136) please use the more common notation 1/sin(x) instead of cosecans csc(x), or define it

------> We added a definition of csc(x) immediately after (136)

15. Figs. 4 and 5.: do you have an intuition why in this particular case only from such a large value of M between 50-100 onwards the approximation works well? This is very much in contrast to few per cent for all other cases, due to the cancellation at d=n?

------> We believe that the main difference between the apparent rate of convergence in Figs. 4 and 5, and, e.g., Figs. 1 and 2 is due to the following reason. When writing the result as a series in N (instead of M), one finds that some of the terms cancel out. Compare e.g. Eqs. (68) and (69), and see the text following (69). As a result, when plotting the results as functions of N as we did in Figs. 1 and 2., the relative size of the correction terms is smaller than when plotting the results as functions of M as we did in Figs 4 and 5. In fact, this also happens in n=d=1, see Eq. (134), which explains why in Fig. 3 the agreement is so good. In addition to this point, the rate of convergence of course also depends on the relative magnitudes of the coefficients in expansions such as (68). For these we do not have any good intuition. (We did not make any changes to the text in connection with this comment).

References: [A] Lewin, M. (2022). Coulomb and Riesz gases: The known and the unknown. Journal of Mathematical Physics, 63(6).

Additional changes to the manuscript

  • We changed K_mu to K_N in Eqs. (84) and (92) for consistency of notation

  • We updated Refs. [43,54] from the arXiv version to published version.

Docutils System Messages

---

## Editorial Decision

published